


# Integrating social, economic, and environmental risk into flood management of aging dam infrastructure by combining cost-benefit and multi-criteria decision analyses

Cyndi V. Castro[1], Hanadi S. Rifai[1]

[1]Civil and Environmental Engineering, University of Houston, Houston, Texas, USA

*Correspondence to*: Hanadi S. Rifai (rifai@uh.edu)

**Abstract.** Management planning for aging dam infrastructure is typically conducted through the lens of a traditional cost-benefit analysis, in which flood characteristics are related to implementation costs while lacking endogenous consideration of environmental risks (i.e., pollutant dispersion, habitat disruption) and social impacts (i.e., vulnerability, community buy-in,

hazard resiliency). To address this gap, we integrate cost-benefit ratios into a spatial multi-criterion decision analysis framework that amalgamates a suite of social and environmental criteria with stakeholder-defined weights and inundation outputs from standard flood control modelling. We use this framework to assess the costs and trade-offs for eight (8) alternative mitigation strategies associated with the Addicks and Barker Reservoir System in Houston, Texas, USA under extreme rainfall conditions. This case study illustrates how the total effectiveness of flood management scenarios may shift when flood

modelling outputs are combined with spatially distributed environmental and social risks. We merge quantitative and qualitative data for high-risk decision-making, thereby fostering stakeholder collaboration amongst conflicting goals.

## 1 Introduction

In the United States alone, there are over 90,000 artificial dams, including various flood control reservoirs, recreational lakes, water supply resources, and hydropower facilities, many of which were constructed with earthen materials following the U.S.

Flood Control Act of 1936. Urbanization and climate change has amplified the water pressure within these aging structures, thereby increasing the risks for significant dam failure. Over one-third of the nation's dams have been classified as 'Significant-Hazard Potential', 'High-Hazard Potential', or completely 'Deficient', according to the level of structural integrity and the severity of consequences in the event of a breach (ASCE, 2017); thus, the impaired infrastructure systems must be strategically managed to reduce the risk of widespread flooding.

The severity of this issue was highlighted during Hurricane Harvey when two high-hazard flood control reservoirs, namely, the Addicks and Barker Reservoir System (ABRS) in Houston, Texas, USA, were challenged by unprecedented amounts of rainfall. Typically, flood control reservoirs mitigate risk by storing large volumes of stormwater and systematically releasing flows through timed operations to minimize downstream impacts. However, the storage load during Hurricane Harvey posed a risk of catastrophic failure for the ABRS, which could have resulted in a wall of water damaging much of the metropolitan



area. Instead, the dam operators chose to release the stored water at emergency-induced surcharged levels, thereby inundating the communities immediately downstream of the reservoirs for several weeks (USACE, 2017). In addition to damaging several thousand businesses and homes, the floodwaters triggered distribution of various chemicals and contaminants (Kiaghadi and Rifai, 2019), which disproportionately impacted low-income and minority populations (Bodenreider et al., 2019).

As observed during Hurricane Harvey, aging dam infrastructure is not equipped to handle intense increases in rainfall, and
emergency conditions may quickly arise. Dam mitigation measures designed to restrain large volumes of water are often prohibitively expensive (Sung et al., 2018). For this reason, flood risk management has trended toward a synergistic approach that combines hard-scale structural measures with softer adaptation approaches. For example, the latest planning frameworks for the ABRS included engineered systems (i.e., additional reservoir storage, levees, tunnels, channel improvements) coupled with non-structural solutions (i.e., community buyouts, optimized timing of releases, flood warning systems) (USACE, 2020).
In determining which measure(s) to choose from a variety of alternatives, traditional flood management employs a cost benefit analysis (CBA) that ranks the reduction in flood risk as a function of overall cost. Comprehensive impacts, including environmental contamination (i.e., toxic soils, superfund sites, industrial pollution) and social vulnerabilities (i.e., poverty, disabilities, language barriers), are known to influence total flood risk but are considered secondary in management frameworks. Once a project has been selected on the basis of CBA, detailed environmental and communal investigations are
then pursued. Nonetheless, local conditions interact with hydrologic phenomena and must be unequivocally represented within decision-making frameworks for a truly resilient approach to flood risk management.

Here, we combine economic, hydrologic, environmental, and social considerations into a composite risk framework for enhanced decision-making by considering the case study of the ABRS under Hurricane Harvey rainfall conditions and various alternative mitigation strategies. Our proposed workflow merges socio-demographic and environmental characteristics with
watershed modeling through a multi-criterion decision analysis (MCDA), which we then integrate into a standard CBA framework. We consider the case study of reservoir-induced flooding during Hurricane Harvey as an opportunity to further investigate complexities associated with dam management and how these processes impact the surrounding community during extreme event conditions. Unique hydrological phenomena, such as cross-basin overflow and emergency-induced reservoir releases, are integrated into the framework to amalgamate synergies associated with complex systems of watersheds, rather
than studying a single watershed in isolation. We extend the popular MCDA approach to not only improve flood control policy associated with dam infrastructure but also to elucidate how complex engineered solutions impact the tripartite coupling of human-water-environmental systems in an urban setting.

In Section 2, the background of the case study is presented, including a review of environmental and social factors that compound flood impacts associated with the ABRS. Here, alternative mitigation strategies that have been proposed by the
U.S. Army Corps of Engineers (USACE) for addressing the aging dams are described. Section 3 explains the methodology for creating hydrologic and hydraulic models, translating risks into spatial indices, and integrating these criteria into a CBA+MCDA framework. Results for the case study are presented in Section 4 and further discussed in Section 5.



## 2 Background

### 2.1 Integrated flood management

Traditional flood management involves the narrowing-down of numerous mitigation options into a select focused array through high-level flood modeling and a standard cost-benefit analysis (CBA) (Brouwer and van Ek, 2004). Social and environmental considerations are loosely considered in such approaches but are not included as dependent variables in the decision-making framework. Instead, a preferred mitigation option is often chosen from the primary perspective of flood inundation extents and economic costs. Detailed site assessments are then performed during the engineering project phase. For example, in the latest

planning study published by the USACE for the ABRS system, economic costs and the extents of flood protection formed the fundamental basis for project recommendations. Environmental resources were introduced qualitatively, with a description of potential impacts to natural habitats. Environmental service facilities (i.e., treatment and industrial facilities) were mentioned only briefly and were largely indeterminate. Socio-demographic factors were also summarized at a high-level for each watershed boundary (population, income, education, and race) (USACE, 2020). However, a quantitative method for including

such considerations into the overall decision-making process and how such risks could influence the proposed alternatives was lacking.

Flood reservoir planning may benefit from the use of a multi-criterion decision analysis (MCDA) approach for evaluating alternatives through explicit consideration of societal and environmental risks. MCDA is a blanket term used to describe a variety of methods that evaluate multiple, and often overlapping, criteria in a structured decision framework (Voogd, 1982).

Common MCDA approaches within the hydrological literature include analytic hierarchy processing (Le Cozannet et al., 2013), fuzzy set theory (Li, 2013), multi-attribute utility theory (Yang et al., 2012), simple additive weighting (Liu et al., 2014; Sjöstrand et al., 2018), evolutionary optimization (Karterakis et al., 2007), and TOPSIS (Lee et al., 2013) (Velasquez and Hester, 2013). Researchers are increasingly trending toward participatory modeling for flood vulnerability assessments and management approaches due to the numerous qualitative benefits gained through stakeholder involvement (de Brito et al.,

2018; Karjalainen et al., 2013; Ronco et al., 2015). Here, we focus on the application of spatial MCDA by employing a simple additive weighting (SAW) approach using stakeholder feedback in a geographic information system (GIS) environment. Spatial MCDAs are used to integrate overlapping criteria with stakeholder values, which have gained popularity with the increased availability of geographic datasets (Malczewski and Jankowski, 2020). GIS-based MCDA tools allow consideration of the tradeoffs involved in different scenarios and visualization of how decisions impact the region.

In the context of flood risk management, spatial MCDAs have been largely used to evaluate the overall net impact of flood magnitude for alternative mitigation measures. The interdisciplinary application of MCDAs for social vulnerabilities and environmental risk assessment has received considerably less attention (de Brito and Evers, 2016; Malczewski, 2006). MCDAs have become increasingly popular in water resources planning and management (de Brito and Evers, 2016; Hajkowicz and Collins, 2007), but such approaches for flood-control dams have primarily focused on optimization of release operations and

not the planning of new structures (i.e., Chu et al., 2015; Fu, 2008; Labadie, 2004; Zamarrón-Mieza et al., 2017). The extension





of MCDA approaches to consider large-scale infrastructure planning for dam systems is a largely undeveloped area of research (de Brito and Evers, 2016; Zamarrón-Mieza, 2017). As we experience a continued shift toward more extreme storm events and aging dam infrastructure, we necessitate a planning framework that evaluates risk comprehensively. Figure 1 describes the traditional decision-making workflow for flood management, as exemplified in the latest USACE planning study for the
ABRS (USACE, 2020), while contrasting our proposed workflow that combines spatial MCDA techniques with CBA and inundation modeling.

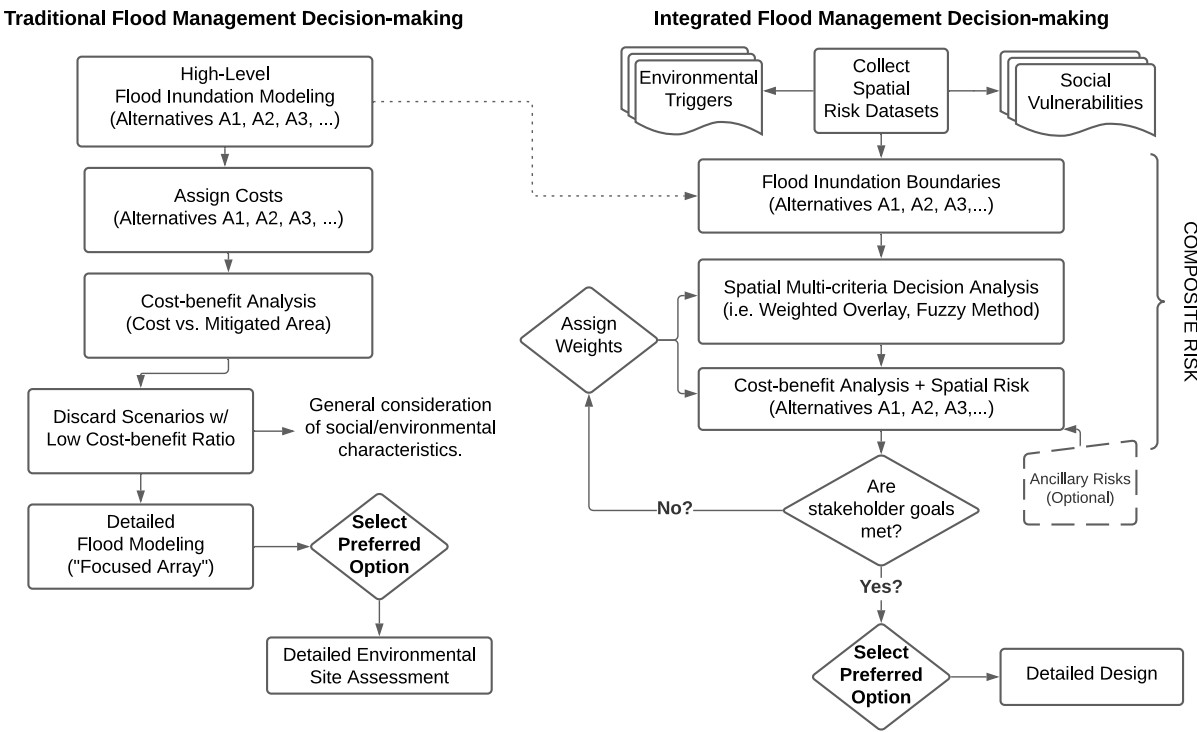

**Figure 1:** Traditional cost-benefit analysis framework compared with the integrated multi-criteria decision analysis framework for decision-making of alternative flood management scenarios.

**2.2 Case Study: The Addicks and Barker Reservoir System**

The ABRS comprises several watersheds in the Houston region that are hydrologically-connected via the Addicks and Barker flood management dams and their downstream releases into Buffalo Bayou, as well as cross-basin overflow from Cypress Creek that enters the reservoir watersheds during extreme events (Fig. A2). The reservoirs are operated by the USACE and have been classified as two of the most-hazardous and deficient dams in the United States (USACE, 2010). During Hurricane
Harvey, large volumes of water spilled over the Cypress-Addicks watershed divide and entered the local reservoirs, introducing significant uncertainty regarding the ability of the reservoirs to withstand the stored pressure (Sebastian et al., 2017). To accommodate these increased inflows, and to reduce the risk of catastrophic dam failure during Hurricane Harvey, the





reservoirs were released according to unprecedented surcharge procedures, causing widespread flooding in the receiving channel and damaging thousands of structures (HCFCD, 2020; USACE, 2017). Simultaneously, overland flow conditions in

the adjacent watersheds interacted with the reservoir releases and compounded regional flood conditions. Flooding associated with Hurricane Harvey damaged over 154,000 homes in the greater-Houston region, of which at least 46,800 flooded structures were located within the ABRS inner-connected watershed system (HCFCD, 2018). The extent of damages inspired widespread discussions regarding regional drainage infrastructure, with specific attention to mitigation of the ABRS reservoirs (USACE, 2020).

**2.2.1 Proposed mitigation strategies**

Following Hurricane Harvey, discussions regarding how best to address the flood risks associated with the ABRS were raised throughout the community. Popular mitigation solutions included dredging a large underground tunnel, adding an additional reservoir to capture cross-basin overflow, widening receiving channels, increasing storage capacity, optimizing release operations, and buying-out properties. Such strategies are reminiscent of the original 1940 ABRS project plan, where additional

open space, reservoir storage, and routing improvements provided an added layer of protection but were later abandoned due to limitations in funding and land availability (Fig. A2; USACE, 1940). In 2018, the Harris County Flood Control District (HCFCD) and the USACE commissioned a joint study of the ABRS system to evaluate various recommended solutions. An interim report was released in October 2020 describing preliminary recommendations for the ABRS strategies. Eight alternative scenarios were screened on the basis of CBA and narrowed to a focused array of five mitigation strategies

recommended for further analysis (USACE, 2020, Table 3). We analyzed these eight preliminary alternatives through our proposed CBA+MCDA framework to investigate the impact of considering comprehensive environmental and social risks in the screening phase, summarized in Table 1. Hydrological assumptions for each of the eight alternatives are described in Section 3.1.

**Table 1:** Array of alternative mitigation strategies and cost estimates, in 2020$ million (M), from the October 2020 U.S. Army Corps of
Engineers interim feasibility report (USACE, 2020). *: The diversion levee for cross-basin overflow was excluded from the USACE report. Instead, we used the monetary value for diversions between adjacent watersheds in the USACE (2020) report (per linear unit) and extrapolated to assumed lengths for this case study.

| Alternative | Description | Cost Estimate |
|:---:|:---|:---:|
| A1 | No action. Model is used for baseline comparisons. | - |
| A2 | Additional reservoir to capture cross-basin overflow. | 2,500 M |
| A3 | Large-scale property buyout plan upstream of reservoirs. | 5,000 M |
| A4 | Diversion levee to re-direct cross-basin overflow. | *1,200 M |
| A5 | Large-scale property buyout plan along receiving channel. | 2,300 M |
| A6 | Increased storage capacity + optimized timing of releases. | 1,600 M |
| A7 | Widen and deepen receiving channel. | 1,100 M |
| A8 | Underground tunnels to re-route existing reservoir outflows. | 9,250 M |





### 2.2.2 Environmental and social impacts

During Hurricane Harvey, highly industrialized regions of Houston were impacted and released various pollutants into the
environment. The floodwaters distributed over one-million gallons of hazardous materials throughout the region (Christine
and Yue Xie, 2018; Miller and Craft, 2018), resulting in widescale and long-term health impacts from exposure to bacteria,
chemical toxins, mold, and carcinogens (Horney et al., 2018; Kapoor et al., 2018; Schwartz et al., 2018; Stone et al., 2019).
Various environmental consequences were associated with ABRS releases, such as flooded wastewater treatment plants,
leaking storage tanks, contaminant dispersal, sediment redeposition, and disturbance of acidic soils (Folabi, 2018; Kighadi and
Rifai, 2019). Research following Hurricane Harvey also highlighted the varied social factors that contributed to disparate flood
impacts and resiliency among population groups. The Houston region lacks formal zoning and is comprised of a highly-
heterogeneous composition of vulnerable populations interspersed with wealthier communities (Christine and Yue Xie, 2018).
Vulnerable populations are at higher risk of experiencing post-traumatic stress disorder and long-term health effects after a
flood disaster and tend to be the slowest to recover, leading to endemic poverty issues (Dickerson, 2017; Grineski et al., 2020).
Wealthy and middle-income populations face higher risks when located outside of federally-designated floodplains where
flood insurance is voluntary (Dickerson, 2017). Mobility issues associated with flooding reduced access to emergency services,
which posed additional hazards to vulnerable populations and led to several fatalities during Hurricane Harvey (Bodenreider
et al, 2019, Chakraborty et al., 2019, Jonkman et al., 2018).

In considering the proposed ABRS mitigation strategy of adding a third reservoir, local concerns have been raised regarding
the environmental disruption of conserved prairie lands that provide natural stormwater mitigation, recreational opportunities
(TPL, 2018), and habitat preservation, including the federally-endangered prairie dawn-flower (FWS, 2021). The proposed
strategy of channelizing the ABRS receiving stream also presents habitat disruption for a highly-threatened species, the
Alligator snapping turtle (Munscher et al., 2020), and other aquatic biota. We noted negative social connotations associated
with channelizing the natural stream, which has long-been opposed by local residents due to recreational and amenity
preferences (Schafer, 2013). We also considered a high societal risk for the mitigation option of diverting cross-basin overflow
to an adjacent watershed, which has been shown to increase community flood risk downstream (GHFMC, 2019). While these
issues have been studied as individual occurrences, there exists a limited understanding of how such factors interact holistically
and impact cross-basin mitigation planning. As such, the practical integration of environmental and social systems into flood
risk management requires further efforts for improved resiliency.

## 3 Methodology

### 3.1 Hydrologic and hydraulic modelling

We performed flood modeling for each of the alternative mitigation strategies by coupling the USACE's Hydrologic Modeling
System (HEC-HMS) and Hydraulic Modeling System (HEC-RAS) (Fig. S1-S3). While the ABRS system is comprised of
several inter-connected watersheds, we limited our mapping extent to the Addicks and Buffalo Bayou Watersheds, since these





areas were most impacted by the array of mitigation strategies. We incorporated hydrological conditions for adjacent watersheds into the Addicks and Buffalo Bayou models, further described in the Appendix.

Baseline HEC-HMS and HEC-RAS models were downloaded from the HCFCD Model and Map Management (M3) System platform (HCFCD, 2019) and applied to Hurricane Harvey rainfall conditions. Multi-sensor, quality-controlled radar and rain gauge data was derived from the National Oceanic and Atmospheric Administration (NOAA) for hourly time-series estimates from August 24, 2017 21:00 to August 29, 2017 23:00 (NOAA, 2017a) (Fig. S4). The baseline models were calibrated to USGS stream gauge flows (USGS, 2019), high water marks (HCFCD, 2017), and high-resolution imagery (NOAA, 2017[b]) during Hurricane Harvey.

HEC-HMS Version 3.5 hydrologic models were created with the HMS-PrePro Toolbox (Castro and Maidment, 2020) using the Curve Number method in the Addicks Watershed (Table S1) and the Green and Ampt method in the Buffalo Bayou watershed (Table S2). Peak flows from the HEC-HMS output hydrographs were used as inputs to the HEC-RAS models (Table S3) for graphical analysis of flood inundation. Hydraulic geometries were obtained from HCFCD M3, and steady-state flow analyses were conducted in HEC-RAS Version 5.0.1 with subcritical flow (HCFCD, 2019). The upstream boundary conditions were modeled with a normal depth slope equal to the average of each stream reach. Downstream boundary conditions were set at critical depth. HEC-RAS Mapper was used to create depth and inundation boundaries (Fig. S3) according to 2018 Harris County LiDAR topography, 10 cm resolution (TNRIS, 2019). Modeling assumptions for each of the alternative scenarios included:

- A1 (Baseline): Models were downloaded from the HCFCD M3 system, updated using latest geospatial datasets, and calibrated to field observations for comparison to alternative mitigation strategies.

- A2 (Additional Reservoir): We assumed the maximum storage volume of an additional reservoir at the Cypress-Addicks drainage divide was $2.34 \times 10^8$ m$^3$ (190,000 acre-foot) with a 56.6 m$^3$/s (2,000 ft$^3$/s) outflow near the Bear Creek tributary, per (HCFCD, 2015 and USACE, 2020); however, our model only captured approximately $1.23 \times 10^8$ m$^3$ of flow (100,000 acre-foot) under Hurricane Harvey conditions (see Appendix Text A2). We linked the Cypress Creek hydrological model with the Addicks model by simulating diversion nodes to capture the estimated quantity of cross-basin overflow (Fig. S5).

- A3 (Addicks Watershed Buyouts): In this scenario, we assumed approximately 10,000 homes located below the Addicks Reservoir spillway are purchased and reallocated. We therefore adjusted the curve number values in each of the impacted subbasins to account for an increase in open space (Table S1).

- A4 (Diversion Levee): We assumed that one-half of the transfer observed during Hurricane Harvey continued as overflow into the Addicks watershed, while the other one-half of overflow was diverted to Cypress Creek by a levee. We incorporated assumptions regarding the risk of increased flooding in the lower portions of Cypress Creek by adopting inundation bounds from GHFMC (2019), which modeled a diversion levee at the Addicks-Cypress Creek watershed divide.

- A5 (Buffalo Bayou Buyouts): We assumed that 441 structures along Buffalo Bayou were acquired in this scenario (per USACE, 2020) and modified the associated subbasin parameters to account for these changes. We removed associated land parcels from the composite social risk maps in this alternative.

- A6 (Increased Storage): Here we assumed that an increased storage capacity within the existing bounds of Addicks and Barker Reservoirs would allow for an optimized release strategy into the receiving channel. We adapted release


rates from the USACE 2010 Interim Control Action Plan (USACE, 2010), where constant releases began once the stream flow at the Piney Point gauge reached 113.27 cms (4,000 cfs) (see Appendix Text A2).

-    A7 (Enlarged Receiving Channel): We modified the Buffalo Bayou channel geometry within the HEC-RAS model to capture an additional 340 m³/s (12,000 ft³/s) capacity. In this scenario, the modified channel cut depth averaged 3.0-4.6 m (10-15 ft) with 1V:4H side slopes, daylighting to natural ground. The proposed top widths in our channel geometry ranged from approximately 100-160 m (350-540 ft).

    -    A8 (Underground Tunnel): Here we assumed no outflows from the Addicks or Barker reservoirs into the receiving
215        channel, thereby removing inflows from the HEC-HMS source nodes for Buffalo Bayou. Per USACE (2020), reservoir storage water would be re-routed around the city and toward Galveston Bay in this mitigation scenario.

## 3.2 Spatial multi-criteria decision analysis

Here, a GIS-based MCDA simple additive weighting (SAW) approach (known as the "weighted overlay" method) was used to spatially link the hydrological and hydraulic modeling with environmental and social flood risk. SAW MCDA approaches
establish value functions by multiplying each criterion by a specific weight and summing the total numerical scores. SAWs have the advantage of being intuitive for decision-makers and have observed wide application throughout the hydrological literature (de Brito and Evers, 2016; Velasquez and Hester, 2013). Typical weighted overlay steps include:

1) Define the problem, goal, or objective holistically.
2) Determine criteria and constraints from local sources and expert opinions.
3) Standardize factors into a common scale through reclassification.
4) Rate and weight the importance of each factor.
5) Aggregate layers and criteria into an overall suitability map.
6) Apply constraints, as applicable.

In *Step 1*, the study objective was defined as identifying the mitigation alternative that minimizes economic, social, and environmental risks while providing maximum flood protection benefits. The contributing criteria (*Step 2*) were determined from a literature search of environmental and social vulnerabilities exacerbated by flooding (Section 2.2), complemented with local knowledge regarding several specific alternatives. For *Step 3*, each of the environmental and social factors were converted to a standardized gridded dataset with a uniform scale from 0 to 100 (where higher numerical values represented greater risk).
To standardize the point and polyline feature classes into spatially varied datasets, the Euclidean Distance method was applied. Euclidean distances convert feature layers into gridded datasets by assigning a value to each cell that indicates the distance of that cell to the nearest criterion, thus standardizing space and creating hotspots in multi-criteria decision making. In *Step 4*, the influence of each criterion was incorporated into the analysis by multiplying the standardized datasets by the weights in Table 2. The weighted layers were aggregated (*Step 5*) to produce composite risk maps within the ABRS system in terms of
environmental and societal criterions and classified into levels from low to high risk. A flood inundation mask was then applied to each composite index to constrain the functions to areas that would be impacted in the modeled scenarios (*Step 6*). Thus, the total risks analyzed for each alternative included a hybrid combination of environmental, hydrological, and societal factors. Composite risk maps were derived by multiplying stakeholder-chosen weights (*w*) by normalized evaluation scores (*e*) for each identified criterion (*j*) using the weighted overlay method, Eq. (1):


$$R_{(E|S)} = \sum_{j=1}^{n} w_j e_j,$$ (1)

Where $R_{E|S}$ refers to the composite risk value of the gridded cells for each spatial map within the domain (E: environmental, S: social), n represents the number of criteria in each domain, $w_j$ refers to the relative weight of each criterion ($j$), and $e_j$ represents the normalized evaluation score (0 to 100).

Composite impact functions were obtained from Eq. (2) for environmental degradation and social effects using zonal statistics

for the composite risk and the modeled inundation area of each alternative ($i$):

$$IF_{(E|S)i} = \frac{\sum(\bar{R}_{(E|S)i}*a_i)}{\sum a_i} + \text{ancillary risk}, \text{ for } i=1,2,\dots,8,$$ (2)

where $IF_{E|S}$ represents the impact function for each domain, $\bar{R}_{E|S}$ describes the average risk value in each zone, and $a_i$ refers to the zonal area impacted in each alternative. We defined ancillary risk as an additional measure of risk value specific to select alternatives, further described in Section 3.2.2.

**3.2.1 Comprehensive risk datasets**

An exploratory geospatial review was conducted to evaluate the applicability of various criterions identified in the literature search, including water quality point-sources, petroleum storage tanks, superfund sites, wastewater treatment plants, petroleum terminals, natural gas facilities, landfills, oil spills, air quality stations, mold growth, medical facilities, population growth, educational centers, emergency response locations, ethnicity, income, transportation, utilities, and damaged structures. These

datasets were consolidated into the primary factors that spatially correlated to changes in flood patterns within the ABRS system, shown in Table 2. Toxic release inventories, leaking petroleum storage tanks, wastewater treatment plants, and the potential for soil erodibility were used to create a composite geospatial map of environmentally-sensitive areas in the reservoir watershed system. Stream samples were obtained from field campaigns following Hurricane Harvey (Kiaghadi and Rifai, 2019), which were used in this study to validate the areas of environmental burdens associated with contamination in local

waterways. The Center for Disease Control (CDC) maintains a Social Vulnerability Index (SoVI) that combines various risk factors related to natural disaster preparation and recovery. The CDC SoVI has been used in various studies as a suitable index for flood hazard mapping due to the underlying criteria, including socioeconomics, household composition, disabilities, minority status, language, housing, and transportation (Flanagan, 2011). An independent analysis validated the CDC SoVI for use in disaster planning and adaptation (Bakkensen et al., 2017). For purposes of this study, the SoVI index was used as the

primary factor in societal risk related to flood hazards. Additional social factors used in the MCDA included population density, flood insurance, roadway inundation, and proximity to medical facilities. The spatial risk associated with flood insurance was derived from national flood hazard zones and a repository of damaged structures in the community. It was assumed that residents within the FEMA 1% and 0.1% flood zones carried flood insurance, while 20% of all other residents had purchased voluntary insurance (Klotzbach et al., 2018). The depth of roadway inundation was chosen as a limiting mobility factor for

access to and from emergency services. Water inundation was exported from pre-defined hydraulic modeling ensembles and used to select roadways that would be inaccessible.


**Table 2.** Data sources and weightings for environmental and social criteria.

| Dataset | Source | Weight |
|---|---|---|
| Toxic Release Inventory | Environmental Protection Agency (EPA, 2016) | 5% |
| Leaking Petroleum Storage Tanks | Texas Commission Environmental Quality (TCEQ, 2019) | 20% |
| Wastewater Treatment Plants | City of Houston (COH, 2019) | 45% |
| Soil Erodibility | U.S. Department of Agriculture (USDA, 2019) | 30% |
| Medical Facilities | City of Houston (COH, 2019) | 5% |
| Population Density | Census Bureau (USCB, 2019) | 20% |
| Inundated Roadway | TIGER and Modeling (USCB, 2019) | 5% |
| Flood Insurance | National Flood Hazard Layer (FEMA, 2019) | 5% |
| Social Vulnerability | Centers for Disease Control (CDC, 2016) | 65% |

### 3.2.2 Ancillary risk datasets

Several mitigation alternatives included ancillary risks for land preservation, biodiversity, and social impacts. Location
assumptions for the additional reservoir and cross-basin overflow patterns were georeferenced from HCFCD (2015), and conserved prairie lands were obtained from TPL (2018). Geospatial bounds for protected lands along Buffalo Bayou were obtained from the United States Geological Survey Protected Areas Database (USGS PAD-US), which represents public lands held for conservation purposes (USGS, 2018). Potential structures for property buyouts were estimated from generalized data points for Hurricane Harvey flood claims from the Federal Emergency Management Association (FEMA, 2017) overlayed
with volunteer HCFCD buyout locations (HCFCD, 2021). We assumed these parcels would be converted to open space land use classification, which was incorporated into the hydrological modeling and risk calculations. We also adopted a flood inundation boundary published by GHFMC (2019) as a conservative proxy for estimating the additional flooded area and social burden incurred by transferring flows from the upper-Addicks watershed to lower portions of Cypress Creek. We overlayed the inundation bounds from GHFMC (2019) with the Hurricane Harvey inundation bounds from HCFCD (2017b)
to estimate the spatial flood difference. For Alternatives A2 and A7, the areas of impacted conservancy lands (prairies and channel easements, Fig. 3) were assigned $\bar{R}_E = 100$ and integrated into the weighted $IF_E$ calculations. Similarly, the composite social risk $IF_S$ for A7 included $\bar{R}_S = 50$ for the natural recreational areas disturbed by channelization. In Alternative A4, the social impact associated with increased flooding along Cypress Creek was assigned $\bar{R}_S = 100$ to quantify the added communal stressors from worsened downstream conditions. Finally, we included $\bar{R}_S = 50$ for the areas of property buyouts along Buffalo
Bayou in Alternative A5, to quantify how this region is comprised of established neighborhoods who have strongly resisted buyout efforts in the past (Houston Strong, 2020).




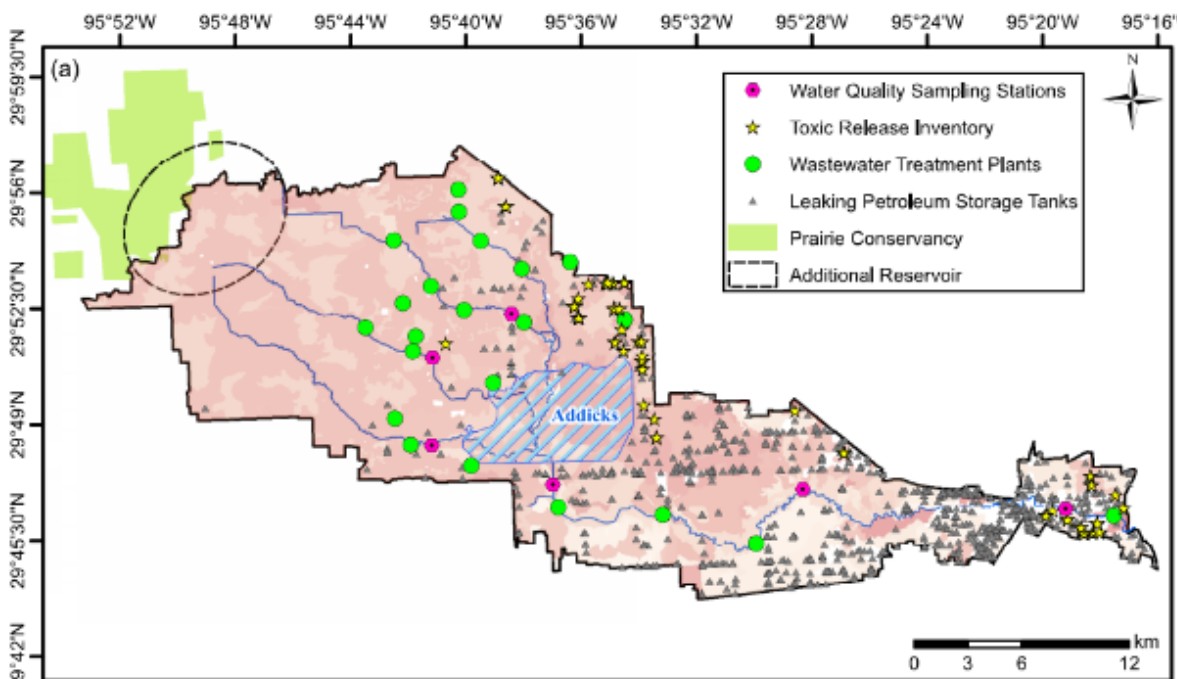

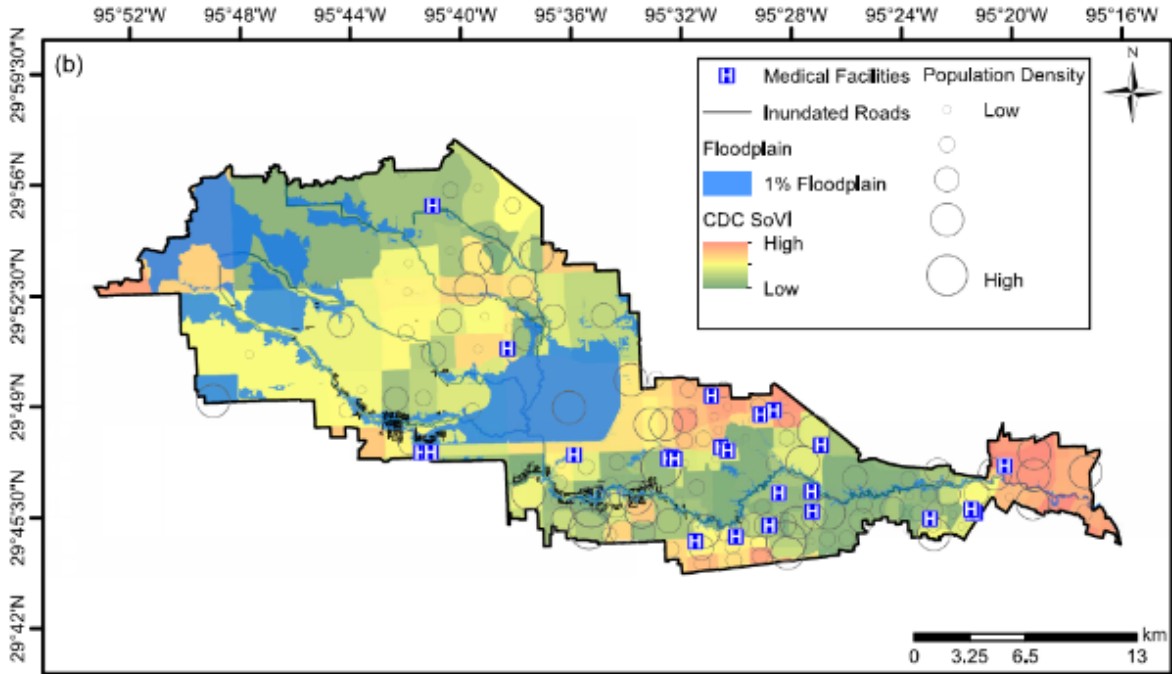

**Figure 2:** Identified causative factors for Addicks-Buffalo Bayou combined watersheds in Houston, Texas, USA during Hurricane Harvey conditions for environmental contamination risks (a) and social vulnerabilities (b).





### 3.2.3 Weight determination

We assigned general weights for each criterion (Table 2) according to discussions with Houston-area flood risk stakeholders, including governmental entities, interest groups, and specialized consulting firms. This knowledge-based approach follows the participatory planning nature commonly associated with MCDA SAW frameworks. In participatory decision-making, external stakeholders collaborate to guide the impact and importance of the multiple criteria used in the MCDA (Mendoza and Martins,

2006). Structured approaches for quantifying the evaluation criteria from stakeholder judgements include the Delphi technique (Lee et al., 2013; Pathak et al., 2020), the step-wise assessment ratio (Khosrayi et al., 2018), focus groups, and cognitive mapping (Mendoza and Martins, 2006). As participatory modelling is inherently qualitative, individual criterion weights will differ according to local conditions and stakeholder goals. We presented the weights in Table 2 as a proof-of-concept for the CBA+MCDA framework; we, therefore, anticipate formal adoption of this framework might explore several of the additional

participatory modeling methods available per local resources and goals. By considering multiple, oftentimes conflicting sets of criteria, participatory weighting coalesces quantitative techniques with qualitative expert opinions to facilitate discussions, elucidate internal values, and aid in justifying final decisions (Mendoza and Martins, 2006).

### 3.3 Cost-benefit analysis

CBA frameworks are used to compare multiple water policy options on the basis of economic efficiency, which is measured

as the difference between added costs and benefits (Ward, 2012). In the context of classic flood management decision-making, added costs include the implementation costs of any new mitigation strategies, and added benefits include a reduction in flood risk when compared to the status quo condition. Additional considerations often include long-term operational and maintenance costs, investment costs, economic growth potential, and non-monetary societal costs (i.e., community willingness, business disruption, housing relocation) (Jonkman et al., 2004). Cost-benefit indices were derived according to Eq. (3) using

the estimated implementation costs for each of the mitigation alternatives (USACE, 2020) and the avoided flood risk area, obtained from hydrologic and hydraulic modeling. Areas were converted to monetary units by averaging the cost per unit area in the traditional CBA calculations (USACE, 2020, Table 7), resulting in a flood inundation conversion factor of 0.478 $M/hectare for this case study.

$$CB_i = \frac{C_i}{B_i}, \text{ for } i=2,3,\ldots,6, \tag{3}$$

where $CB_i$ refers to the cost-benefit indicator, $C_i$ represents the implementation cost in 2020$ million, and $B_i$ represents avoided inundated area (inundated area $A_1$ – inundated area $A_i$), converted to monetary units.



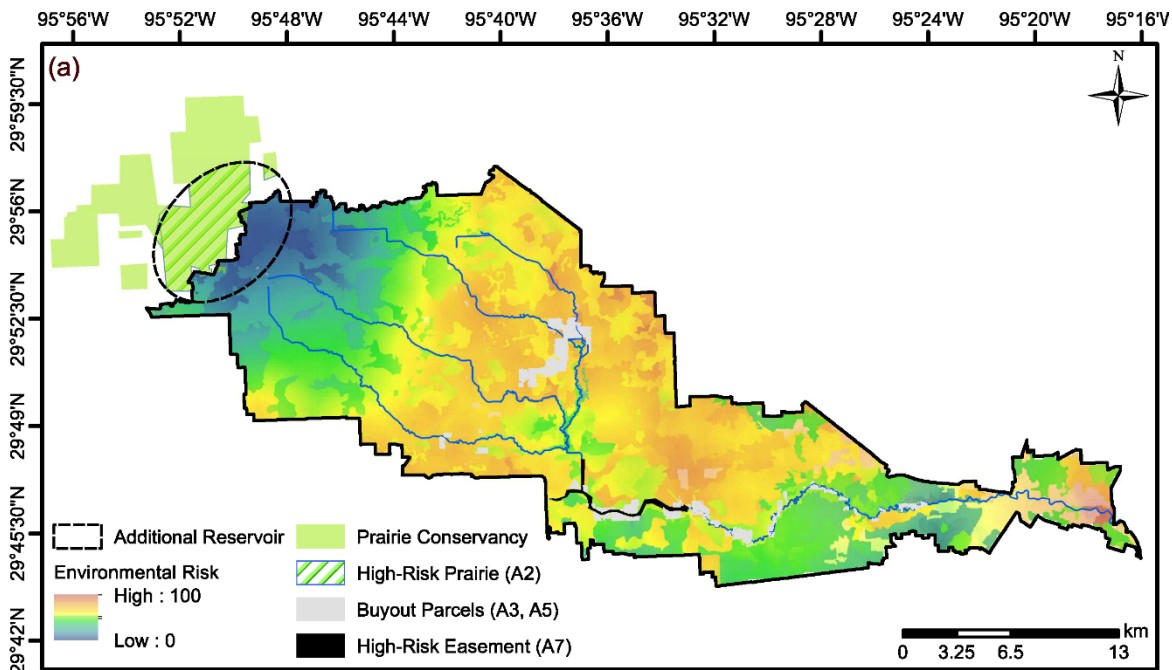

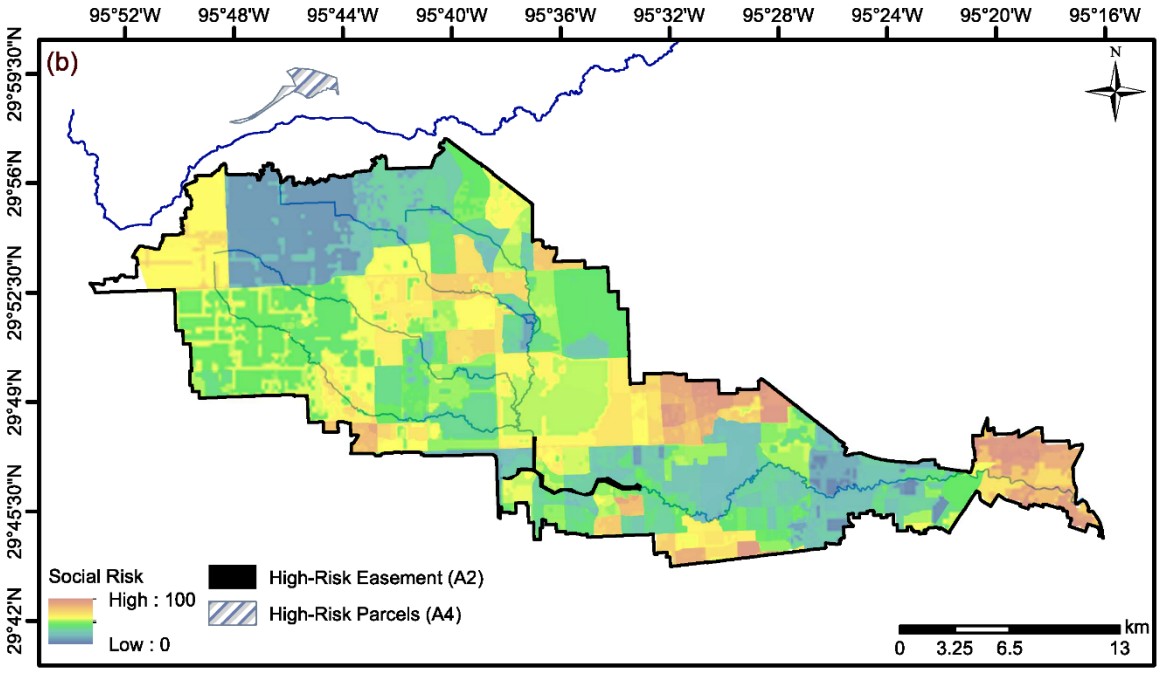

**Figure 3:** Composite risk map for environmental criteria (a) and social factors (b) in the Addicks-Buffalo Bayou combined watershed, Houston, Texas, USA, including ancillary risk for select mitigation alternatives.




## 3.4 Integrated CBA+MCDA

We merged the CBA and MCDA approaches by ranking Alternatives A1-A8 for each set of indicators (CB, $IF_E$, $IF_S$) and assessing the results relative to each reference alternative. Since the unique indicators contained different units of measurement ($/hectares, 0-100 risk) we used z-score normalization to transform the values to equivalent scales. We
aggregated the goals by assigning an equal weight to each of the three dimensions in our framework (economic, environmental, social) and used a linear additive model to integrate the sub-indicators. We assumed mutual, preferential independence between each of the unique sub-indicators for the linear additive aggregation method; however, we understand this approach may not be useful in all mitigation scenarios, and we recommend the analysis team to consider outranking methods and geometric aggregation, as applicable (Nardo et al., 2005). We normalized each set of data using the z-score
methodology and defined the total risk score according to Eq. (4) and Eq. (5), respectively:

$$Z_{(CB|E|S)i} = \frac{indicator(CB_i, IF_{Ei}, IF_{Si}) - mean\,(CB_i, IF_{Ei}, IF_{Si})}{standard\,deviation\,(CB_i, IF_{Ei}, IF_{Si})}, \tag{4}$$

and

$$T_i = Z_{CBi}w_{CB} + Z_{Ei}w_E + Z_{Si}w_S, \tag{5}$$

where $Z_{(CB|E|S)i}$ describes the z-score of the indicator for domain (CB: cost-benefit, E: environmental, S: social), $w_{(CB|E|S)}$ refers
to the weighting of each domain (equal weighting of 1/3 in this case study), and $T_i$ refers to total risk.

## 4 Results

### 4.1 Hydrologic and hydraulic modelling

Hydrologic and hydraulic models were used to obtain flood inundation extents for each of the proposed ABRS mitigation alternatives, A1-A8. Alternatives A1 through A4 corresponded to changes in flood inundation primarily within the Addicks watershed (Fig. 4), and Alternatives A1, A5-A8 represented mitigation options that would impact the Buffalo Bayou watershed
(Fig. 5). Total simulated inundated areas for Alternatives A1-A8 (hectares) were: 6,774.92 ($A1_{Addicks}$), 3,184.14 (A2), 6,308.72 (A3), 5,243.13 (A4), 1,456.45 ($A1_{Buffalo}$), 1,417.31 (A5), 519.58 (A6), 382.59 (A7), 560.61 (A8). We summarized observations regarding the modeling results below:

- A1 (Baseline Conditions): The baseline conditions model for the Addicks watershed compared well against observed
stream gauge heights (Table A2), with an $R^2$ coefficient of 0.97 and a Nash-Sutcliffe efficiency (NSE) of 0.95. The resulting flood inundation bounds also compared well against manual spot inspection of flooded areas, per NOAA aerial imagery (NOAA, 2017b). These results demonstrate the accuracy of the model in capturing overland flow conditions within the watershed and cross-basin overflow from Cypress Creek. The baseline conditions model for the Buffalo Bayou watershed compared well against observed stream gauge heights (Table A2), with an $R^2$ coefficient
of 0.99 and a Nash-Sutcliffe efficiency (NSE) of 0.95. The resulting flood inundation bounds also compared well against manual spot inspection of flooded areas, per NOAA aerial imagery (NOAA, 2017b). These results demonstrate the accuracy of the model in capturing overland flow conditions within the watershed and the observed reservoir releases from Addicks and Barker.
- A2 (Additional Reservoir): As described in Appendix Text A3, we estimated approximately $1.23E^8$ m$^3$ (100,000 acre-
feet) overflowed from the Cypress Creek watershed into the Addicks watershed during Hurricane Harvey, which




corresponded well to observed flooding conditions. An additional reservoir would therefore need to capture at least this amount to minimize cross-basin transfer. The USACE (2020) resiliency study proposed an additional reservoir capacity of 2.34E$^8$ m$^3$ (190,000 acre-feet). Since we assumed a constant outflow from the reservoir into Bear Creek (HCFCD, 2015), the majority of flood inundation improvements between A1 and A2 were observed near this confluence. While the addition of a third reservoir improved conditions, a significant portion of the flows into the Addicks Reservoir resulted from overland conditions within the watershed. By comparing outflow hydrograph volumes from HEC-HMS (Fig. A4), we estimated only 16.42% of the flows into Addicks Reservoir were accounted by the cross-basin transfer, thereby suggesting additional mitigation measures are warranted to limit substantial inflows to the aging dams.


- A3 (Buyouts): When considering the hydrological impacts for buyouts by altering the subbasin loss parameters, we observed negligible changes to the overall peak flow conditions, primarily due to the location of the buyout parcels near the downstream bounds of the model. Since peak flows were used to drive the HEC-RAS models, the inundated areas for A3 were nearly identical to baseline conditions, suggesting an unreasonable improvement to flood conditions given the high estimated cost ($5B).


- A4 (Diversion Levee): While the USACE (2020) mitigation report excluded diversions from the focused array due to the adverse flood impacts in adjacent watersheds, we chose to simulate model conditions for purposes of comparison in the CBA+MCDA framework. We demonstrated a significant reduction in flood inundation area within the Addicks watershed in comparison to baseline conditions (6,774.92 to 4,734.85 hectares). However, these benefits were partially by additional flooded area in the downstream portions of Cypress Creek. These findings highlight the hydrologically-interconnected nature of cross-basin dams and stress the necessity to consider social impacts as a function of regional space.



- A5 (Buyouts): Modelling results for Alternative A5 were similar to observations for buyout conditions in the Addicks watershed. Specifically, the peak flow outputs differed a negligible amount when altering the subbasin loss parameters due to the location of the parcels near the receiving stream. We therefore removed the area of buyout parcels (in both A3 and A5) from the inundated area to represent limited flood impacts in these regions.


- A6 (Increased Storage Capacity): By increasing the storage capacity in the existing Addicks and Barker dams, we assumed optimized timing of reservoir releases into the receiving channel. Alternative A6 resulted in a significant reduction of flood inundation area by approximately 64% from baseline conditions (1,456.45 to 519.58 hectares). We noted the USACE (2020) mitigation study did not inherently link the provision of optimized releases by increasing storage capacity. Our findings suggest that additional capacity in the existing reservoirs may have alleviated the need to release surcharged flows into the receiving channel during Hurricane Harvey by allowing stored floodwaters to remain within the reservoirs for a longer period of time (Fig. A3).


- A7 (Enlarged Receiving Channel): In order to achieve adequate storage of the observed releases during Hurricane Harvey, we needed to expand the top width geometry for Buffalo Bayou in our HEC-RAS model to extents much wider than what was proposed by the USACE (~100-160 m in our model vs. 70 m in USACE, 2020). While our study is not intended for detailed design, significant displacement of land would be necessary for this alternative scenario, impacting social and environmental considerations along the banks of this natural stream.


- A8 (Underground Tunnel): We noted similar improvements to the inundated area for Alternative A8 (560.61 hectares) when compared with A6 (519.58 hectares), highlighting how the receiving channel is driven primarily by outflows from the dams. The significant costs associated with the tunnels ($6.5-12B) and the comparable flood mitigation benefits to other alternatives led the USACE to drop this option from the focused recommendations (USACE, 2020); however, as described in Section 4.2, when we consider the added risks and benefits associated with social and environmental impacts, the tunnel alternative warrants further consideration.




**Figure 4:** Flood inundation bounds for the Addicks watershed (a) and Buffalo Bayou watershed (b), simulated with Hurricane Harvey precipitation conditions in HEC-HMS and HEC-RAS.



## 4.2 Integrated risk analysis

By using the traditional CBA framework for the dam mitigation options, Alternatives A2 (Additional Reservoir) and A7 (Channel Improvements) resulted in the lowest $CB_i$ ratios for the Addicks and Buffalo Bayou watersheds, respectively. As such, the USACE (2020) interim study prioritized these strategies in a recommended focused array of optimal plans (A4 was not considered in the interim study). Alternative A5 (Buffalo Buyouts) was included in the proposed focused array only as a means for constructing the widened receiving channel (USACE, 2020). Shortly after publication of the interim dam study, a

report was compiled by a local coalition of flood resiliency stakeholders highlighting the need for further attention to ecological and social factors associated with the ABRS mitigation alternatives. These constituents urged the USACE to further explore Alternatives A6 (Increased Storage) and A8 (Underground Tunnels) in light of holistic environmental and social considerations. This report also stressed the negative social impacts associated with parcel buyouts for the socio-demographics represented along Buffalo Bayou, particularly if the buyouts are conducted in a patchwork manner (Houston Stronger, 2020).

In echoing these concerns, our study demonstrates how quantitative inclusion of social and environmental criteria can alter the ranking of potential flood mitigation strategies. Table 3 summarizes the results of the case study calculations, Fig. 5 shows changes to the rankings between the CBA and the CBA+MCDA frameworks, and Fig. 6 compares these integrated outputs across the financial, social, and environmental domains.

**Table 3:** Results of the integrated cost-benefit analysis and multi-criteria decision analysis for composite risk of Alternatives A1-A8 for the ABRS in the Addicks ($_A$) and Buffalo Bayou ($_B$) watersheds.

| | Alt. | Description | $C_i$ | $A_i$ | $CB_i$ | $IF_{Si}$ | $IF_{Ei}$ | $Z_{CBi}$ | $Z_{Si}$ | $Z_{Ei}$ | $T_i$ |
|---|---|---|---|---|---|---|---|---|---|---|---|
| **Addicks** | A1$_A$ | Baseline Model | $ 0 | 6,774.92 | 100.0 | 38.19 | 78.65 | 0.92 | -0.26 | -0.14 | 0.17 |
| | A2 | Additional Reservoir | $ 2,500 | 3,184.14 | 31.34 | 37.02 | 84.37 | -0.97 | -0.74 | 1.38 | -0.11 |
| | A3 | Property Buyout | $ 5,000 | 6,308.72 | 95.54 | 37.67 | 78.25 | 0.80 | -0.47 | -0.24 | 0.03 |
| | A4 | Diversion Levee | $ 1,200 | 5,243.13 | 38.98 | 42.46 | 75.37 | -0.76 | 1.47 | -1.01 | -0.10 |
| **Buffalo Bayou** | A1$_B$ | Baseline Model | $ 0 | 1,456.45 | 100.0 | 38.23 | 75.51 | 0.81 | 0.12 | 0.19 | 0.37 |
| | A5 | Property Buyout | $ 2,300 | 1,417.31 | 99.19 | 42.01 | 63.98 | 0.77 | 1.04 | -1.66 | 0.05 |
| | A6 | Increased Storage | $ 1,600 | 519.58 | 72.01 | 32.22 | 75.83 | -0.57 | -1.35 | 0.24 | -0.55 |
| | A7 | Channel Improvements | $ 1,100 | 382.59 | 53.34 | 41.15 | 81.01 | -1.49 | 0.83 | 1.07 | 0.14 |
| | A8 | Underground Tunnels | $ 9,250 | 560.61 | 92.86 | 35.18 | 75.37 | 0.46 | -0.63 | 0.16 | 0.00 |



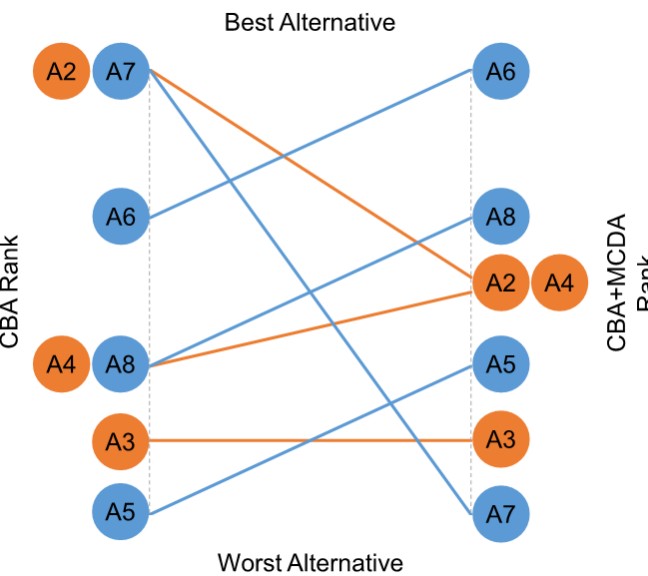

**Figure 5:** Ranking of alternatives using the traditional CBA framework compared with the integrated CBA+MCDA approach.

In comparing the $CB_i$ ratios for each of the alternatives, we see that the buyout alternatives (A3 and A5) provided the lowest benefit when viewed strictly in terms of flood benefits due to the hydrological properties of the watersheds, which resulted in a minimal reduction of inundation risk (Section 4.1). However, when we integrated environmental and social considerations among the alternatives, we noted a potential for tradeoffs in cumulative risk (i.e., A5 surpassed the rank of A7 due to the high environmental and social consequences associated with channelization). In the Addicks watershed, our composite risk framework maintained the disinclination toward A3 by incorporating high costs of buyouts with a minimal improvement in social or environmental conditions. We noted a reduction in the relative preference of A2 (Additional Reservoir) when compared with A4 (Diversion Levee) for mitigating cross-basin overflow, with the $T_i$ for each alternative nearly identical. If we were stakeholders during the planning of this case study, such a finding would encourage us to consider coupled flood modeling for the Cypress Watershed prior to discounting A4. By investigating the resulting composite risk maps for A2, which was a preferred mitigation strategy

in the USACE (2020) report, we noted the flood risk was diverted largely to unpopulated, low-vulnerability areas with higher likelihood for soil erosion potential, thus lowering the choice suitability. Moreover, as discussed in Section 4.1, the addition of a third reservoir here does not fully mitigate the flood issues with the Addicks watershed, which is driven largely by overland flow. By collectively viewing the synergies between these domains, we identified a need to further investigate alternative strategies for mitigating the cross-basin overflow.

In the Buffalo Bayou watershed, the preferred mitigation option in the CBA framework (A7 – Channel Improvements) transitioned to the highest-risk alternative when considering CBA+MCDA. Alternative A7 would require substantial changes to Buffalo Bayou, which is highly opposed by residents, without necessarily providing additional improvements in comparison to other mitigation options presented here. Alternative A6 (Increased Storage) resulted in the most preferable option due to less relative ancillary risks than A7 and substantial flood reduction. Moreover, Alternative A6 provided opportunity for

optimized timing of reservoir releases, an inherent benefit that was not considered in the USACE (2020) approach. The next highest-ranked scenario in the Buffalo Bayou cohort was Alternative A8 (Underground Tunnels), which was initially discarded in the CBA framework due to the significantly high costs associated with subsurface construction. When incorporating the reduced environmental and social impacts conferred by re-routing flood water away from the urban centre, Alternative A8 became a higher-performing option, warranting inclusion in the focused array.



**Figure 6:** Standardized z-score for the Addicks watershed (a) and Buffalo Bayou watershed (b) using the combined CBA+MCDA framework. Spider graph showing performance of each alternative within the economic, social, and environmental domains for Addicks watershed (c) and Buffalo Bayou watershed (d).



Due to the compensatory nature of the linear aggregation methods employed in this study, the comprehensive risks and subsequent rankings of alternatives will be sensitive to changes in the stakeholder-defined weights (i.e., Table 2, Eq. 5). We suggest that a high sensitivity of weightings is advantageous at the screening stage of flood management, particularly when conflicting goals are present across diverse domains. This method of aggregation reflects the multiplicity of stakeholders' viewpoints and fosters an iterative and interactive approach to scenario-based management. By coupling the framework results directly with stakeholder values, we facilitate the need for greater cooperation among various interest groups. For example, following the ABRS interim study (USACE, 2020), several community groups had expressed concern regarding the CBA-based recommendations due to conflicting goals stemming from unique perspectives and expectations for the mitigation alternatives (Houston Stronger, 2020). When we integrated environmental and social factors into our CBA+MCDA framework, we noted a change in alternative rankings, which could be used to foster further discussion between the interest groups. This benefit to participatory modelling facilitates transparency regarding goal magnitudes, which helps decision-makers gain a greater understanding of their own assumptions and values (Mendoza and Martins, 2006). The acceptability of trade-offs between diverse criteria becomes easier to identify and justify, and stakeholders are provided a clear opportunity for active engagement in the decision-making process. The CBA+MCDA framework can then be performed in an iterative fashion by altering stakeholder weights and assessing outcomes until the communal goals of the stakeholder cohort is satisfactory (ref. Fig. 1). This iterative approach allows us to explore the direct feedback of human valuations on the interconnected environmental, social, and hydrological system and how perceptions of different domains may impact the choice of local mitigation strategy (Khan et al., 2017).

## 5 Discussion

The case study of the Addicks and Barker reservoir system highlights the potential risk and complex interactions involving flood control dams. Here, we examined ongoing mitigation strategies associated with dam structures that had been classified as highly hazardous and had been challenged by an unprecedented amount of rainfall during Hurricane Harvey. The 90,000 dams within the United States have an average age of 60+ years (USACE, 2017). These aging structures, as well as the hundreds of thousands of dams throughout the world, risk structural failure and subsequent catastrophic flooding in light of increased climate intensification. We necessitate a streamlined approach to considering environmental contamination, habitat disruption, social vulnerability, and other stakeholder-driven factors when choosing how best to mitigate aging dams, especially considering the substantial costs associated with new infrastructure and cascading social/environmental effects.

By incorporating the inter-disciplinary science of hydrological modeling with social vulnerability and environmental risk, it is possible to consider conflicting demands and tradeoffs across the flood control domain. The CBA+MCDA method presented in this paper may be used to integrate robust stormwater modeling scenarios with multiple risk factors to better understand the correlation of hydrologic systems with overall vulnerabilities. Such comprehensive indicators of risk provide insight into the regional effects of large-scale mitigation decisions regarding extreme storm events. By linking traditional hydrological





modeling with GIS-based risk assessments, the differential impacts of flooding on a population can be analyzed over space for informed mitigation strategies. This approach to dam management and planning integrates the decision-maker as an explicit endogenous driver to the holistic human-water-environment system in response to increasingly complex storms, societies, and environments. When taken, such approaches will enhance our ability to form actionable insights regarding community

resiliency.

Here we used the GIS-based SAW MCDA approach as a proof of concept to showcase how considering multiple spatial criteria within a flood management framework can improve stakeholder visualization and discussion of mitigation options that may have been discarded if viewed strictly through the lens of CBA. We encourage future research regarding additional MCDA approaches, methods for assigning stakeholder weights, and location-specific criterions to better understand the level of detail

necessary for adequately framing the discussion and ensuring the integration of environmental and social domains in flood risk management workflows. While the operational manuals for the studied reservoir system contained guidance for emergency-induced surcharge releases (USACE, 2012), such drastic measures had never before been necessary prior to the unprecedented rainfall observed during Hurricane Harvey. As climate change continues to stress aging dam structures, and as populations continue to densify around urban centers, we anticipate that typical operating procedures for flood control dams

will become increasingly challenged. We, therefore, must consider both the soft approaches and the traditional hard-scale engineering solutions for dam management, which will require an extension of the CBA paradigm to consider both the humans being impacted by the proposed alternatives and also the environments in which the systems reside.



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

**Supplementary materials:** There are no additional supplementary materials.

**Author contribution:** Hanadi Rifai and Cyndi Castro conceived the idea and research approach. Cyndi Castro prepared the manuscript, developed the model, and performed the analysis. Hanadi Rifai advised on content and edited the written paper.

**Competing interests:** The authors declare no competing interests.

**Acknowledgements:** The National Science Foundation grant #1840607 to Hanadi Rifai funded the research.



## Appendix

### Text A1 – History of the Addicks & Barker Reservoir System.

The ABRS has experienced a long history of flood management issues. After two devastating floods in 1929 and 1935, the Addicks and Barker Reservoirs were authorized under the Rivers and Harbors Act, later modified by the U.S. Congress Flood Control Act of 1939 (Cotter and Rael, 2015), to provide protection to Houston's Downtown district and the Houston Ship Channel. The original 1940 project plan included three reservoirs (Addicks, Barker, and White Oak, shown in main text, Fig. 2) with diversion levees and canals to prevent overflow from Cypress Creek and to convey releases around Houston toward Galveston Bay (USACE, 1940). The

Addicks and Barker reservoirs were constructed from 1942-1948, which, at the time, were approximately 25 kilometers west of the Houston city limits in largely unpopulated prairie lands (Wurbs, 2000). Land development quickly spread to the protected areas throughout the 1950s, and the remaining items from the original plan were eliminated (additional reservoir, diversion channels), due in part to rising land costs and availability of space (Rivera Ramirez, 2004). As demonstrated by the timeline in Fig. 3, several major rain events occurred throughout the decades following construction of the reservoirs, prompting ongoing concerns regarding the

ABRS system capacity. Throughout the 1970s-2000s, major subdivisions were constructed within the limits of the reservoir pool levels, raising the risks of flood damage if the reservoirs were to fill at maximum capacity. Prior to Hurricane Harvey, rain events had not directly stressed the ABRS watersheds to the point of triggering emergency-induced surcharge releases, but ongoing reservoir warning reports highlighted the significant impacts of such a risk occurring in the near future (HCFCD, 1994; HCFCD, 2015; USACE, 2008). Several failure zones developed in the earthen reservoir outlets, prompting classification of the reservoirs' safety

rating to Level I: Urgent and Compelling in 2010. A Level 1 classification suggests that without intervention, the dams are "almost certain to fail under normal operating conditions from immediately to within a few years" (USACE, 2010). Shortly after the dams were re-classified, studies emerged warning of the ability of the reservoirs to withstand further increases in climate change and land development (Sass, 2011). The reservoirs encountered several 500-year storm events in succession (2015-2016), triggering record cross-basin overflow conditions and maximum pool levels in Addicks and Barker (HCFCD, 2016; HCFCD, 2018). Plans were

proposed for structural improvement of the aging reservoirs (USACE, 2012a; USACE, 2013); however, many of the modifications were large-scale in nature and had not been completed at the time of Hurricane Harvey.

### Text A2 – Addicks & Barker Reservoir Release Operations.

The optimal release of flood control reservoirs is a primary factor involved in mitigating flood risk, however, there remains significant uncertainty regarding how such release schedules should be crafted and executed (Rivera Ramirez, 2004). Uncertainty in reservoir

releases stems from the imprecise science of estimating available storage capacities, rainfall conditions, and inflow volumes through simulation models. A degree of uncertainty in large-scale reservoir releases is generally acceptable under average rainfall conditions, since "normal" operating conditions limit the amount of damage allowed in the downstream receiving channel while reducing overall flooding. Under extreme stormwater conditions, however, emergency-induced surcharge releases may be triggered that are intended to reduce the risk of dam failure and spillage by potentially and drastically exceeding downstream channel capacity (Rivera Ramirez,

2004). In the ABRS system, total combined releases are typically determined according to peak flows at the United States Geological Survey (USGS) Piney Point stream gauge along Buffalo Bayou (ref. main text, *Fig. 2*). Normal operating procedures for the ABRS traditionally limited releases to 2,000 CFS at the Piney Point gauge to control downstream flooding (USACE, 2009; USACE, 2012b). After a national risk assessment was conducted for the ABRS in 2010, an Interim Reservoir Control Action Plan was developed that increased allowable standard releases from 2,000 CFS to 4,000 CFS, as necessary, to reduce pressure on the dams (USACE, 2010).

Prior to Hurricane Harvey, the Interim Reservoir Control Action Plan releases had only been used once (Tax Day Flood of 2016), which successfully restored the reservoir holding capacities while minimizing risk to downstream property owners (HCFCD, 2016). During Hurricane Harvey, the pool levels in the reservoirs had surpassed critical levels (USACE, 2017), and floodwaters in the reservoirs were released according to an emergency-induced surcharge schedule (USACE, 2012b). Reservoir releases observed during Hurricane Harvey were obtained from USACE Press Releases (USACE, 2017), as shown in Table A1, and incorporated into

the hydrological models. Available data suggested that 13,300 cfs combined was released from Addicks and Barker into downstream Buffalo Bayou. By simulating these flow conditions, the modeled water surface elevations (WSEL) matched observed high-water marks (OHWM) from field inspections and aerial imagery (HCFCD, 2017a; NOAA, 2017b). Therefore, the Hurricane Harvey release values shown in *SI Table 1* were assumed reliable for purposes of this study. The optimized timing release schedule (for modeling Alternative 6) was derived by simulating the overland flow conditions within the Buffalo Bayou watershed using Hurricane Harvey

rainfall in HEC-HMS. The model was then used to determine when the USGS Piney Point gauge would have reached 4,000 cfs (per the Interim Reservoir Control Action Plan), assuming the reservoirs contained adequate storage capacity during Hurricane Harvey.





**Text A3 – Cross-basin Overflow from Cypress Creek.**

The Addicks, Barker, and Cypress Creek hydrological models were linked by simulating diversion nodes in HEC-HMS for cross-basin overflow (Fig. S1, S2). Diversion tables for the Cypress Creek overflow were obtained from local basin models and used as
source gauges in the adjacent watersheds (Fig. S5) . Cross-basin overflow from Cypress Creek to the reservoirs was estimated to be 92,000 ac-ft, based on HEC-HMS model simulations. Observed high-water marks from HCFCD (2017a) were also used to estimate the overflow volume for comparison. The water surface elevation was interpolated from high water marks and intersected with the underlying digital elevation model for an estimated overflow volume of 115,000 ac-ft. For purposes of this study, the cross-basin overflow volume during Hurricane Harvey was assumed to be an average of 100,000 ac-ft, with approximately 97% of the volume
entering the Addicks watershed, and the remaining flow diverting through Barker and into the Addicks watershed (HCFCD, 2015).

**Table A2:** Simulated reservoir releases for Addicks HEC-HMS model under Hurricane Harvey conditions for observed releases and assumed optimized-timing releases, per USACE (2010).

| Time | Addicks Releases (Harvey) | Barker Releases (Harvey) | Optimized Timing Releases, Combined (A6) |
|---|---|---|---|
| August 24, 2017 21:00 | 0 CFS | 0 CFS | 0 CFS |
| August 28, 2017 04:00 | 800 CFS | 800 CFS | 0 CFS |
| August 28, 2017 12:00 | 3800 CFS | 3500 CFS | 0 CFS |
| August 29, 2017 04:00 | 7300 CFS | 6000 CFS | 0 CFS |
| August 29, 2017 09:00 | 7300 CFS | 6000 CFS | 4000 CFS |
| September 3, 2017 00:00 | 2000 CFS | 2000 CFS | 4000 CFS |


**Table A2:** Observed and modeled water surface elevations for Addicks watershed and Buffalo Bayou watershed HEC-RAS models. All elevations are in datum NAVD 1988.

| Creek Name | Gauge Name | HEC-RAS XS | Peak Observed Elevation (ft) | Peak Modeled Elevation (ft) |
|---|---|---|---|---|
| Mayde Creek | HCFCD Site 2190 | 67829.7 | 144.61 | 145.29 |
| | HCFCD Site 2150 | 33134.2 | 114.95 | 115.58 |
| | USGS Site 08072680 | 28295.0 | 114.72 | 114.97 |
| Bear Creek | HCFCD Site 2180 | 63028.6 | 149.54 | 149.63 |
| | USGS Site 08072730 | 27754.8 | 114.71 | 115.05 |
| Langham Creek | HCFCD Site 2140 | 57359.7 | 134.5 | 135.64 |
| | HCFCD Site 2120 | 33728.9 | 111.4 | 111.07 |
| | USGS Site 08072760 | 33859.9 | 111.85 | 111.78 |
| Horsepen Creek | HCFCD Site 2130 | 14080.0 | 115.9 | 115.92 |


| Creek Name | Gauge Name | HEC-RAS XS | Peak Observed Elevation (ft) | Peak Modeled Elevation (ft) |
|---|---|---|---|---|
| Buffalo Bayou | USGS Site 08073500 | 232632.3 | 77.45 | 77.01 |
| | USGS Site 08073600 | 214953.1 | 71.23 | 71.53 |
| | USGS Site 08073700 | 196463.1 | 63.94 | 62.4 |
| | HCFCD Site 2260 | 184862.8 | 60.30 | 58.03 |




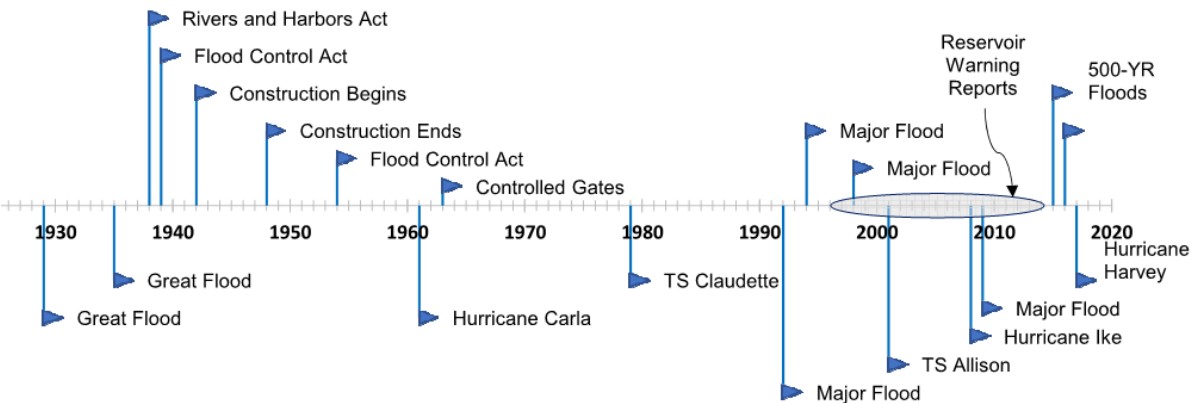

**Figure A1:** General timeline of Addicks and Barker Reservoir construction and major storm events, interspersed with warning reports highlighting the risks of the dams overtopping and/or necessitating emergency-induced surcharge conditions into the receiving channel
(HCFCD, 1994; HCFCD, 2015; USACE, 2008).

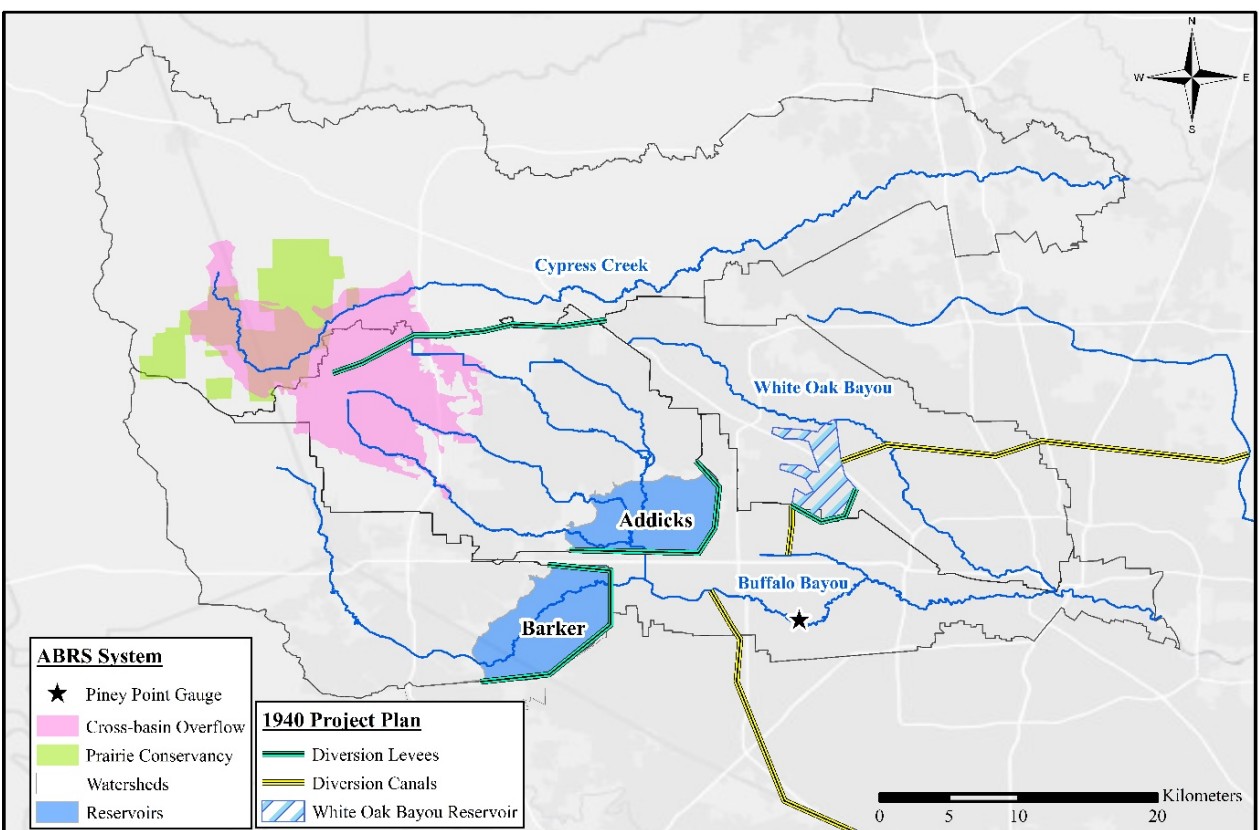

**Figure A2:** Addicks and Barker Reservoir System (ABRS) of regional inter-connected watersheds in Houston, Texas, USA, including components of the original 1940 project plan that were later discarded (USACE, 1940).




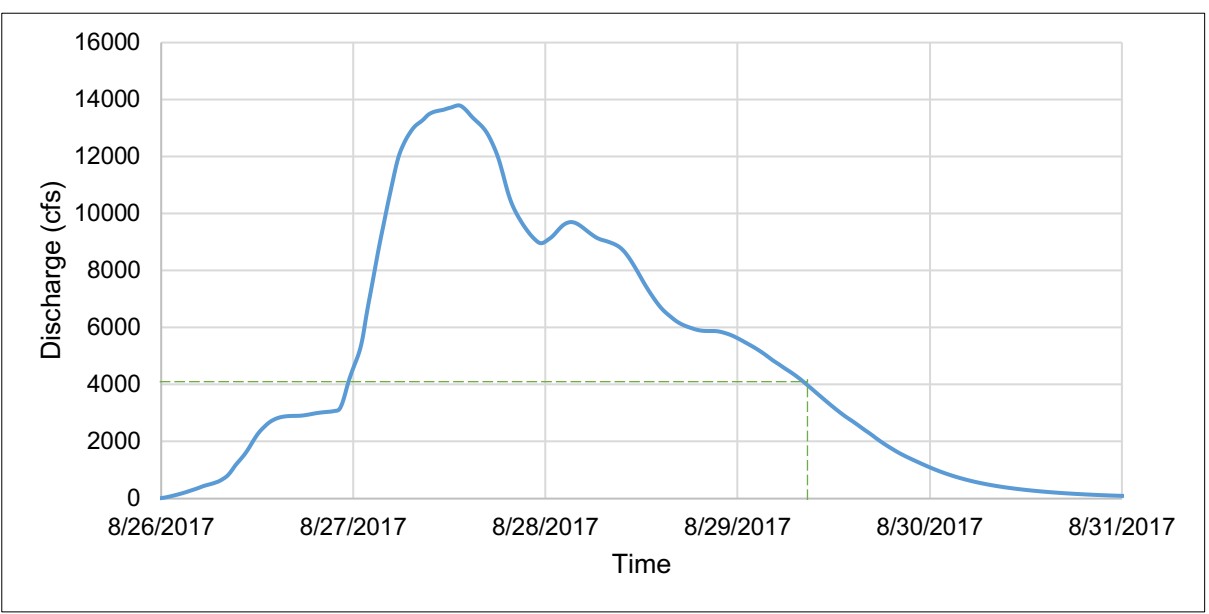

**Figure A3:** Output hydrograph of USGS Piney Point gauge (HEC-HMS Node W1000000_1985_J) under Hurricane Harvey rainfall conditions with no reservoir releases, used to determine the assumed timing of releases for Alternative 6 under the Interim Reservoir Control Action Plan conditions (USACE, 2010).

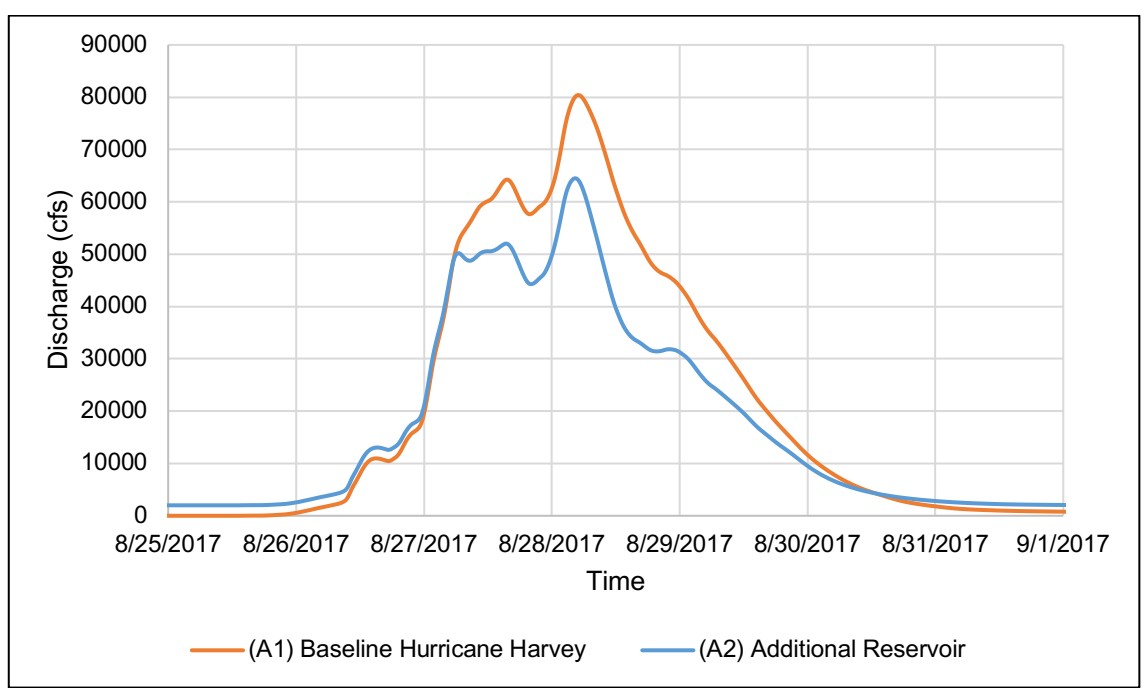

**Figure A4:** Comparison of HEC-HMS output hydrographs at Addicks Reservoir (HEC-HMS node U1000000_9901_J) for Alternatives A1 and A2. The storage volume for A1 is 309,870.1 ac-ft, while the hydrograph volume for A2 is 258,989.6, thereby elucidating the difference in total inflow volume at Addicks Reservoir given the addition of a hypothetical third reservoir.