# Peer review of "Integrating social, economic, and environmental risk into flood management of aging dam infrastructure by combining cost-benefit and multi-criteria decision analyses"

_Natural Hazards and Earth System Sciences, 2021_

## Author Comment (AC1)

**Integrating social, economic, and environmental risk into flood management of aging dam infrastructure by combining cost-benefit and multi-criteria decision analyses**

**Response To Comments – RC1**
**Castro and Rifai**

C#: Comment No., R#: Response No.

C1:     The entire paper, especially the methods section, could be reduced by 20-25% without losing relevant information.

R1:     We have modified the Methods section (and actually, the entire flow of the paper by moving pertinent paragraphs) to improve clarity, reduce extraneous text, and highlight the key takeaway messages, per below comments, specifically: C4, C12, C19, C26, and C37. At least 20-25% of the Methods text was moved to the Supplementary Information document, as suggested, and the entire paper reads more coherently.

C2:     Another criticism is regarding the discussion section, which should be improved to add the limitations of the study.

R2:     We have added a section for Limitations of the study in the revised manuscript by incorporating the below comments, specifically: C3, C13-14, C16-17, C24, and C30-31. We have also strengthened the Discussion section through additional texts that describe the high-risks of reservoir flooding associated with intertwining hydrological, population, and environmental dynamics. We have further strengthened the sections describing how MCDA coupled with CBA allows us to investigate the system as a complex whole and to improve decision-making in light of significant amounts of aging dam infrastructure, urbanization, and climate intensification for rainfall events.

**Specific Comments:**

C3:     A main problem is that the reasoning for model assumptions is not clearly stated (e.g. for the weighting of the criteria, selection of criteria, as well as the points listed in lines 187 to 216). It is not clear how many stakeholders participated, for instance.

R3:     Please see response to C30 where we discuss this in further detail. There was not a formal stakeholder participation modeling exercise conducted (i.e., causal loop diagram, group model-building), but the weights were rather derived from local expertise knowledge according to ongoing relationships and opportunities to work within the pertinent industries (i.e., engineering consulting flood modeling, municipal government flood recovery and sustainability, various U.S. Army Corps of Engineers studies, including Addicks and Barker Reservoirs modeling, academic field investigations and papers describing these reservoirs and their environmental and social impacts throughout the region; see Kiaghadi and Rifai (2019), for example). In practice, if a geospatial overlay approach were used by decision-makers, such as the U.S. Army Corps of Engineers or one of their sub-consultants, their "weighting" valuation would mostly likely be conducted in an iterative fashion, whereby several leaders on the project would define the weights with their counterparts through group discussions, review the results, and then optimize the weights after seeing the results to consider all stakeholder

inputs. This type of an approach allows the decision-makers to reveal their own inherent mental mapping of values and what these results may contribute to the local mitigation options. This would encourage the decision-making team to revise the weightings, if necessary, depending on previous results, and so-on, until they achieve an acceptable balance of local risk (comprehensive hydrological, environmental, and social) and cost-benefits (flood inundation, construction costs, and maintenance costs). We do not anticipate [all] practicing entities to conduct robust participatory modeling techniques prior to each flood management decision (i.e., fuzzy modeling), but we do realize the value in such techniques and have recommended them in the Limitations section for further consideration in future studies.

We discuss in the Limitations section how each stakeholder group would naturally present with varied weighting values, according to local interests, and describe how implementation of this framework must consider this tendency and work toward revealing inherent mental values to facilitation a discussion amongst stakeholders, rather than a defined, one-size-fits-all approach to defining aggregated group weights and selecting the "best" alternative.

C4: The methodology section reads too long and it is a mix of literature review and methods. It should be shortened. It is also not linear and difficult to follow. The authors first describe the weighting procedure and only then detail the criteria used. After that, the weighting procedure is explained in detail. I understand that section 3.2 now tries to add an overview, but it is confusing because many details are not explained. Hence, I would suggest to follow a linear description and incorporate the lines 230 to 255 in the other subsections. This way you will avoid repetition.

R4: Agreed, we have revised Methodology text to include the pertinent methods for the integrated CBA and MCDA approach. The detailed hydrological modeling components where transferred to the Supplementary Information document. We also clarified the weighting procedure, per below comments (C12, C19, C26, and C37).

C5: Similarly, the results section is wordy. The last paragraphs should be included in the discussion. Furthermore, limitations and future work section should be added. Below you can find some suggestions on this (e.g. lack of sensitivity and uncertainty analyses).

R5: Agreed, we have moved the paragraphs into the suggestion sections and have added a Limitations section, per below comments (i.e., C13, C14, C16, C17, C37, C41, and C43).

C6: Line 40: I would say that MCDM is also a tool for traditional flood management. What is innovative of your research is combining both. I would reformulate this paragraph, stating the advantages/disadvantages of each approach, and how their integrated use can provide better answers for an adequate flood risk management. Here you also need to show previous literature that has also followed a similar approach. Try to find flood-related articles that combine both approaches. If they do not exist, you can also list this as an innovation from your paper. These could be relevant articles, but you have to check if they fit the scope as they are not for FRM. https://www.sciencedirect.com/science/article/pii/S2352146515002197 https://www.sciencedirect.com/science/article/pii/S221204161630420X https://link.springer.com/article/10.1007/s40070-019-00098-1

R6: This paragraph has been refined as suggested, including referenced applicable papers utilizing MCDA or CBA and a discussion of integrating both MCDA and CBA into one framework.

We do, however, maintain that MCDA has not been practiced as a flood management approach in recent large-scale dam mitigation studies, at least within the United States (e.g., https://www.nwo.usace.army.mil/Missions/Civil-Works/Planning/Planning-Projects/Cherry-Creek-DSMS/, https://usace.contentdm.oclc.org/utils/getfile/collection/p16021coll7/id/17692, as well as the Addicks and Barker study used for this paper: www.swg.usace.army.mil/Portals/26/BBTnT_Interim_Report_202001001_Final_1.pdf). As such, we describe how MCDA has been used in the academic literature but encourage throughout this paper the adoption of MCDA coupled with CBA as a tool within practicing engineering and decision-making of large-scale dam mitigation studies.

C7:     Line 43: "considered secondary in management frameworks" I disagree that this information is considered secondary. There are hundreds of flood vulnerability studies that show otherwise.

R7:     We appreciate your comment that many academic studies consider this information as of primary importance. In our experience with numerous practitioners within the United States, and per the referenced reservoir management study described in this study, we note that social and environmental impacts are only qualitatively considered, whereas more detailed modeling efforts are devoted to considering the flood inundation conditions when ranking alternatives.

[See, for example, the USACE (2020) Buffalo Bayou Interim Report, https://www.swg.usace.army.mil/Portals/26/BBTnT_Interim_Report_202001001_Final_1.pdf, Sections 2.5, 2.8, 3.2, and 4.10; while socio-demographic and environmental considerations were described qualitatively in this study, a numerical and/or spatial representation of such adverse impacts was not included in the final ranking of alternatives. Rather, the cost-benefit analysis using flood inundation area was compared with the implementation cost as the primary decision-making criteria].

We also realize this consideration is constantly evolving and differs between geographic locations. We intend for this statement to recognize the discrepancy between studies and widespread practice with an encouragement toward explicit quantification and consideration of social/environmental factors within mitigation frameworks. This statement is further clarified in the first paragraph of Section 2.1.

C8:     Line 85-86: This should be in the methods section.

R8:     Text moved to Methodology section.

C9:     Line 94-95: This is an important gap you are helping to fill. This should be mentioned in the introduction section.

R9:     Agreed, moved and emphasized in Introduction section.

C10:    Table 1: It is not clear how you extrapolated the cost estimate. I suggest adding a new column to the table where you can summarize the impacts you describe in Section 2.2.2.

R10:    Derivation of cost estimate from USACE reports was clarified in the revised text. Impacts from Section 2.2.2 were summarized and added to Table 1, as suggested.

C11:    Line 163-166: This is also a gap. I suggest mentioning it in the introduction.

R11:    Agreed, moved and emphasized in Introduction section and the Abstract.

C12:    Line 195: how did you arrive at this number of 10,000 houses? Please detail it more. The same goes for all other quantitative assumptions.

R12:    We gathered this assumption from the referenced documentation (i.e., USACE, 2020) that listed the number of homes assumed by the U.S. Army Corps of Engineers in their study and definition of Reservoir technological alternatives. This text section (per response to C19) was moved to the Supplementary Information document to improve readability and avoid unnecessary details that take away from the key manuscript message while continuing to provide adequate technical detail for reproducibility of the flood inundation models. We have added a reference to the USACE (2020) study in this text location, as well as other quantitative texts throughout this paragraph, for clarity.

C13:    Line 233: How exactly did you determine the relevant criteria? Was there a systematic procedure in the literature review you conducted? This is a gap that should be listed in the discussion section as different scientists would choose different criteria leading to completely different outcomes. Also, what is this "local knowledge"? Did you consult experts in the field? Or it was based on the author's opinion. This should be clarified.

R13:    Clarified in Limitations section, as well as below responses to C14, C16, C17, and C30.

C14:    Line 239: You need to explain how these weights were defined. How many stakeholders were involved? How were they selected? Where do they work and what is their expertise? If they were based only on the opinion of the authors, this should be stated. Furthermore, this should be added as a limitation in the discussion section.

R14:    Agreed, please reference response to C30, where we have described this being based on our expertise from ongoing relationships with local stakeholders. We have further described this in the Limitations section.

C15:    Line 256: What do you mean by "exploratory geospatial review"?

R15:    Further clarified/described in this paragraph.

C16:    Line 260: By consolidated, do you mean you aggregated several criteria into one? If yes, which and how? You should be clearer on the method used to combine these criteria.

R16:    The choice of language here is misleading, as we did not perform a detailed methodological approach to consolidating the datasets chosen, but rather an exploratory investigation into what types of datasets existed in several of the available geospatial repositories (local data sources as well as widespread publicly-available data sources, as referenced in the Manuscript). In future studies, each entity will likely have a personalized set of geospatial datasets, typically hosted on a local server, of which they are most familiar as pertaining to reservoir-induced risk. Further text was added in the Limitations section to describe how the choice of datasets, accessibility, pre-processing, etc. is of vital importance to being able to properly use this type of spatially-based framework, and we necessitate further research into the field of curating and connecting decision-makers with reliable geospatial datasets (the authors are involved in other manuscript preparation efforts addressing this precise need within the literature and industry).

For clarity, we have removed this sentence and have maintained the additional language (Response to C15) to describe how we rather conducted an exploratory geospatial review.

C17:    Line 270: Doesn't the SoVI includes already population density? Wouldn't there be then a redundancy? Ideally you should conduct a PCA or other data reduction techniques. See this article, it may be helpful for the discussion section: https://nhess.copernicus.org/articles/21/1513/2021/.

R17:    The CDC's SoVI does not incorporate Population Density but only 15 census variables (https://www.atsdr.cdc.gov/placeandhealth/svi/documentation/pdf/SVI2018Documentation-H.pdf) at the census-tract scale. Therefore, we noted several areas in our study that appeared to have a "high" social vulnerability risk per the SoVI, but we knew from local experience that these regions were not highly populated (i.e. farmland) and would not pose substantial risk of property damage or loss of life in the event of flooding. We therefore chose to add Population Density as an additional dataset to address the potential of skewing the MCDA approach away from regions where persons reside. This, of course, was particular to our case study region in Houston, Texas, USA and may not be the case in other geographic regions; hence, our emphasis on the types of data layers chosen should be customized to each locale and type of flood management application being analyzed.

        We have highlighted the importance of choosing geospatial datasets according to local expertise, needs, stakeholder goals, and type of study in the Limitations section.

C18:    Line 272: It was assumed based on what? On the information provided by Klotzbach et al?

R18:    Yes, per study conducted by Klotzbach et al. (2018). We updated this sentence to clarify.

C19:    Line 355: The validation against the stream gauge heights is not mentioned in the methods section. Also, why have you conducted validation for some alternatives and not for some? The validation procedure should be described in the methods.

R19:    Comparison against stream gauge heights was mentioned in Methodology, Section 3.1, Lines 176-177. This type of calibration technique is common in industry, but is also not necessary to detail for purposes of describing this framework (since the hydrological modeling component is well-established). In this light, we agree that this is too much detail/text for the reader to digest and takes away from the main message of the paper. Thus, we moved many of the details regarding hydraulic and hydrologic modeling to the Supplementary Information document and reference briefly in the main manuscript text.

**Technical Corrections:**

C20:    Line 9: As a non-native speaker, I had to google what "community buy-in" means. It may be my ignorance, but perhaps you could just frame it as "community acceptance and support" or something similar? Still regarding to this, I do not understand why buy-in and resilience are social impacts. For me they are actually the opposite. I would keep "vulnerability" and use other examples here.

R20:    Agreed. All factors used here to describe social impacts were, in some form, vulnerabilities. We therefore removed the text "community buy-in, hazard resiliency" from this section and further clarified what we meant by both social and environmental impact factors.

C21:    Line 12: remove the (8).

R21:    Text removed.

C22:    Line 84: Remove the word "qualitative".

R22:    Text removed.

C23:    Line 154: please write "third reservoir (A2, Table 1), so the readers can understand that this is one of the 8 alternatives.

R23:    Text added.

C24:    Table 2: you should add the spatial resolution of these data.

R24:    Many of the datasets listed here are point-features, therefore they will not have an inherent spatial resolution. Since we intended for this method to be applicable across geographic regions, we do not want to limit the reader to considering only specific data sources with certain resolutions as applicable to the framework. These data sets were available at the time of analysis and will likely change in the future, according to locale/stakeholder-goals.

        We recognize this is an important point to be clarified and considered; therefore, we have added additional text regarding the choice in geospatial data sets to the Limitations section.

C25:    Line 255: What do you understand by "comprehensive risk dataset" and "ancillary risk datasets"? The difference between the two should be introduced.

R25:    Difference between these two terminologies has been clarified in revised text.

C26:    Line 290: I am not sure, but perhaps you can make a table with this information? Right now the text is too dense and difficult to have an overview of the many assumptions.

R26:    This information has been further refined to describe how we chose average weightings for ancillary risks on a scale from 0 to 100% and to reduce the wordiness of this section.

C27:    Figure 2: The figures have a very low resolution. For the final version please use a pdf or similar graphs.

R27:    High-resolution PDFs were included during the submission process, but the NHESS Preprints incorporate embedded Word (.PNG) images. We will ensure the high-resolution PDF images are used in the final, type-set Manuscript.

C28:    Line 300: The information regarding the weighting should come before.

R28:    Text moved to correspond closer to and reference the weightings identified in Table 2.

C29:   Line 301: Remove "general".

R29:   Removed.

C30:   Line 301: How exactly were these "discussions"? How many stakeholders? How did you achieved consensus between these stakeholders? Was one weighting derived for each participant and then you made an average? The procedure should be clarified.

R30:   This is based on ongoing relationships with various stakeholder entities, including consulting firms, environmental advocates, municipal leaders, and personal knowledge working in these fields over many years that had culminated into the chosen weights. We only included them here as a general idea of how this procedure could be used to quantify environmental/social considerations in such a framework (and for reproducibility). We added additional descriptions in the following sentences to describe how our methodology was a knowledge-based approach; however, more structured participatory modeling/stakeholder-derived weighting approaches can and should be pursued if employing CBA+MCDA in practice.

We have further described this limitation in the "Limitations" section and have revised this paragraph text to make clear how our weightings were not derived in a structured approach, but rather were derived here from the authors' personal culminated knowledge in flood management to showcase how the framework could be used by practicing entities. We do not suggest this study as proposing optimized engineering rankings for the mitigation alternatives but rather as a facilitation tool to foster discussion and analysis of values. For formal weighting in a case study such as ABRS, numerous stakeholders would need to be involved across varying domains, scales, and jurisdictional boundaries, which was beyond the scope of this paper.

C31:   Line 305-307: If I understood correctly, you have not done this. Hence, it should be removed from the methods section. I would add this to the discussion, saying what future research could do/limitations in your study.

R31:   Clarified and moved to Limitations section.

C32:   Line 307-308: This should be in the discussion section.

R32:   Clarified and moved, per C31.

C33:   Line 310-312: This is literature review, not methods… I would remove all together.

R33:   Text removed.

C34:   Figure 3: The color of the high risk easement should be changed as it is now the same as the color used for the study area border.

R34:   Figure 3: Color modified.

C35:   Line 339: The normalization is mentioned 2 times in this paragraph.

R35:   Sentence removed, and Eq. (4) and Eq. (5) referenced further up in paragraph.

C36:   Line 354: Please provide this information in a table format. This way it is easier for the readers to compare the different alternatives.

R36:   Table added.

C37:   Line 354 to 364: The text reads too long and should be cut.

R37:   We suggest maintaining this information in order to address any questions by readers regarding the calibration and reliability of these HEC-HMS hydrological models, as this set of inter-connected watersheds has been notoriously difficult to model in the past (per working within this industry). However, we have moved the group text to the Supplementary Information document, as the emphasis of reliable baseline model results corresponds directly with the discussion of model results for Alternatives A2-A8 in the following bullet points. $R^2$ and Nash-Sutcliffe efficiencies are common metrics used to quickly assess model reliability in comparison to observed values (in academia); further, a comparison of spatial flood inundation bounds (via high water marks and/or spatial imagery) is common for model calibration within industry.

C38:   Table 3: Please add to the legend of the figure what $C_i$, $A_i$, $CB_i$, etc. mean. It is easier for the readers not to need to search back in the text.

R38:   Table 3: Description of variable nomenclature added to Caption.

C39:   Figure 5: I like the figure as it summarizes the outcomes and is easy to understand. However, I do not understand why some alternatives are in orange and some in blue. Please add this information to the legend.

R39:   Figure 5: Legend added to clarify that Orange represents Addicks Watershed Alternatives (A2, A3, A4), and Blue represents Buffalo Bayou Watershed Alternatives (A5, A6, A7, A8).

C40:   Figure 6: very important figure, but difficult to read because is twisted. Please use portrait orientation. Also, add the legend to the y axis. What do high and low z scores represent? Low z scores represent low social risk, for instance?

R40:   Figure 6: Portrait orientation provided. Legend updated to showcase inside of graph. Also, y-axis text updated for clarification about z-scores. Further clarity about z-scores added to caption.

C41:   Line 469-486: This is discussion, not results.

R41:   Text moved to Discussion, Section 5.

C43:   Line 506: Please add a section called conclusion and add the text from here there.

R43:   Section heading added.

---

## Author Comment (AC2)

Comment:      The manuscript presents a MCA analysis of different flood mitigation options for a complex basin in Texas, USA. Topic is interesting and fits the scope of the journal. The idea itself is (obviously) not new in the literature, nor relevant methodological issues are here developed, but the case study is interesting and potentially deserves for publication, once a series of points are modified and/or clarified.

**Response:**     While we appreciate the reviewer's familiarity with MCDA approaches within the academic literature, we similarly acknowledge that such concepts are not employed in-practice within large-scale reservoir planning and mitigation studies, as demonstrated in this article (also see https://www.nwo.usace.army.mil/Missions/Civil-Works/Planning/Planning-Projects/Cherry-Creek-DSMS/, https://usace.contentdm.oclc.org/utils/getfile/collection/p16021coll7/id/17692). Instead, a qualitative description of environmental and societal risks is included in dam planning studies, while the ranking of alternatives during the preliminary planning phase is conducted with cost-benefit analyses based on extent of flood inundation and capital costs. Our goal with this study was not to present a case study to show how MCDA is used but to demonstrate how current practical planning paradigms could be improved with geospatial datasets and overlays in a manner that is intuitive and able to be streamlined into standard modeling approaches (e.g., HEC-HMS/HEC-RAS are the most commonly used programs in the United States for studying hydrology and flood inundation) by combining CBA with MCDA. The integration of CBA with MCDA is novel within the dam planning literature, as most academic studies that used methods from MCDA have focused on reservoir operations optimization and not large-scale planning. This novelty has been further explained and highlighted in the revised Manuscript, Lines 90-99, with additional background literature for support. Here, we have also further identified the novelty of this study that incorporates inter-disciplinary social and environmental considerations within water resources management MCDA approaches for dams, rather than being limited to flood characteristics.

We also realize this consideration is constantly evolving and differs between geographic locations. We intended for this statement to recognize the discrepancy between studies and widespread practice with an encouragement toward explicit quantification of, and therefore integrated consideration of, social/environmental factors within dam mitigation frameworks.

**Major Points:**

*Equation (1) – Composite Risk*

Factors R (Environmental, Social) are weighted averages of evaluation scores. Concept is clear, but we have no information on how such scores are given, we only know about their general meaning (sources of contaminants, soil erodibility, medical facilities…). Lines 256-261 provide a long list of items to be considered in the risk evaluation. However, it is not clear which of them has been really considered in the environmental and social criteria, and how.

Response:     Reference response to comment below for Section 3.2.3.

In different words: what do factors "$e_j$" exactly represent? Are they binary quantities (e.g.: presence of a source of contaminant in a cell), extensive quantities (e.g.: length of inundated road in a cell), intensive quantities (percentage of flood insurances among residents)? How the scoring 0-100 is attributed to each factor for each cell? Do cells have a uniform extension?

Response:        Updated. Reference discussion in Lines 258-272.

In particular "Stream samples were obtained from field campaigns following Hurricane Harvey, which were used in this study to validate the areas of environmental burdens associated with contamination in local waterways." I have not understood how such data were used to define values for the environmental factors in each cell.

Response:        Language updated for clarity in Lines 298-300.

Finally, I strongly suggest providing a short description of the SoVI (Social Vulnerability Index) and variables involved in the index.

Response:        Added, Lines 303-306.

*Equation (2) – Impact Functions*

1) What are the "zonal statistics for the composite risk and the modeled inundation area of each alternative"?

Response:        Clarity added that this is an ArcGIS command, Line 281.

2) What are the "zones"? the inundated areas for each scenario? I understand that "$a_i$" are the corresponding inundated surfaces, is this correct? Or do they also comprehend the areas impacted by the "ancillary risk" (see below)?

Response:        Clarity added in Line 284.

3) What is the summation index in equation (2)?

At the end of the story, I cannot understand IF. If "Rbar" is an average (I guess, spatial average) over the zone and "a" is the area of the zone, than Rbar is constant over the zone and IF=Rbar+ancillary risk. But this has no sense, therefore I conclude that I was not able to understand equation (2).

Response:        IF represents a composite "Impact Factor" for environmental or societal considerations within the watershed of interest.

                 Yes, Rbar is a spatial average of each zone, clarified in Lines 283-284.

                 Ancillary impact was calculated similarly to the composite risk calculation and added, per Eqn. 3. For example, let's look at the Environmental Impact Factor ($IF_E$) for Alternative A2 (Enlarged Receiving Channel). Here, our Zonal average of the environmental risk raster (see Fig. 4a) under the modeled inundation bound for this alternative (see Fig. 5b) equaled 68.12 (for an inundation area of 382.8 he). Then, we assigned a negative environmental impact of 100 (the worst score possible) for this alternative, due to its impact on the highly-threatened wildlife here (in a hypothetical stakeholder situation, one might definitely argue this risk is not "that important", and that group of decision-makers would then have to explain to the ecosystem protection persons why this risk value should be reduced, but as it stands in current planning, while the Alligator snapping turtle may be noted by the USACE in their reports, this

consideration is not explicitly included within the modeling paradigm as a value that can be assigned a number and incorporated into the ranking framework). In assigning this easement area (which was calculated to be 259.99 he in GIS) a risk/impact of 100, we can perform a simple weighting calculation as such:

$$IF_E = \frac{(68.12 * 382.8he + 100 * 259.99he)}{(382.8he + 259.99he)} = 81.01 \ (as\ shown\ in\ Table\ 4)$$

Additional clarification has been added to Eqn. 3 and Lines 283-287, 330-337.

*Section 3.2.2 - Ancillary Risk*

I see conceptual inconsistencies here. Soil use / buyouts due to mitigation measures are not risks, are deterministic impacts; they have 100% probability along the lifetime of the mitigation measure, and they last for all such time. On the other hand, the damage from a flood scenario has a probability of exceedance less than 100% in the given period. What is here called "ancillary risk" is not a risk at all and should not simply be added to the flood risk by assigning certain values of R-scoring for the areas impacted by the measures, as here done.

On the other hand, the extra-flooding expected along the Cypress Creek is an additional/ancillary risk, and it is correct to handle it as such. Why is such area simply accounted by a $R_S$=100 scoring? It should be added to the flooded area and evaluated with respect to CB and IFs.

Response:        Language for ancillary "risks" updated to state ancillary "impacts" throughout manuscript.

The additional flooding along Cypress Creek will remain "ancillary" for purposes of demonstrating our proposed framework, as we aim to keep the three primary domains distinct within the first round of MCDA (environmental and social risks in the area overlayed with flood inundation from modeling within each watershed). The additional "risk" here is not necessarily about the hydrologic inundation but more-so how it will impact the people in this area – in a detailed engineering study, yes, this area would be bound by the flood model. But we are trying to show here that when we model watersheds, which is what is done in practice, there are impacts elsewhere that involve society, and thus should be considered. We could have chosen to represent this ancillary "impact" in the Cypress watershed by the number of persons impacted by transferring flooding elsewhere – for purposes of ease, we chose to use the area of flood boundary. These inter-acting watersheds are particularly complex from a hydrological stance, due to the way water transverses over the watershed divide in certain storm events (this is not typical of most watershed divides). A fully coupled model was outside the scope of this study, and therefore we did not feel comfortable adding the Cypress Creek inundation bounds as part of our inundation area ($B_i$) since we did not perform extensive robust modeling here but rather made a very rough estimate according to inundation bound graphics from another study (Dunbar et al., 2019 – see Supplementary Information).

*Section 3.2.3 – Weight Determination*

This is THE key point of any MCA, making the difference between an exercise and a relevant field case. I cannot really understand how the authors determined the weights. I read about "discussions with Houston-area flood risk stakeholders, including governmental entities, interest groups, and specialized consulting firms"; the description continues with principles derived from the literature (lines 302-307); authors conclude by saying that "As participatory modelling is inherently qualitative, individual criterion weights will differ according to local conditions and stakeholder goals". All this is true, but what did they really do? From lines 308-312 I may understand that weights in table 2 are somehow just a reasonable proposal, not yet evaluated by stakeholders?

It is also very important to clarify that weights are not general but linked to the definition (and consequent variation ranges) for the indicators to be weighted. All this information set should be part of the discussion (with stakeholders) devoted to fixing the weights.

Again, weights are the key point. Selection criteria should be discussed. If such criteria are not robust, an extensive sensitivity analysis should be provided in order to give real value to the MCA.

Response:    Our goal here was not to derive the final weights for optimizing the Addicks & Barker Reservoir system, as we posit that such weighting values cannot be finalized in such a manner since they include issues like social norms, vulnerability, and environmental harm, which all have many societal and diverse responses associated with each. Rather, we are attempting to showcase to stakeholders how a participatory planning approach can be used with MCDA – we are envisioning the spatial MCDA being a quick and intuitive visual manner to assess how if, for example, Stakeholder A really thought that social vulnerability should be given higher weight, while Stakeholder B really though that flood risk should be the only consideration, how when we consider each person/group's values, the outcomes shift. That was the point of the paper, to simply showcase this shift, and suggest that it should be included in the decision-making environment. As such, we learned from much of the social-sciences literature approaches that use weightings as proxies for values, eliciting them through models, and spurring very important discussions at the planning stage so that all stakeholder input can be considered and the decisions to weight X over Y, or vice-versa, has real implications. Hence, we maintain that the weighting here should definitely be subjective, as that is our goal.

To clarify this point, we added significant text to the Revised Manuscript (e.g., Lines 264-269) as well as a Limitations section (Section 5) explaining this proof-of-concept and how more robust weighting methods can be used, if the parties opt to do so, but we did not envision that being the case in a real-world application of our framework.

We selected the criteria based on local knowledge of the many issues that were impacted during Hurricane Harvey flooding. This will very likely be unique in each locale. A sensitivity analysis would showcase which criteria were sensitive to changes to the model but would not help with identifying which ones should be included in the first place. We posit that if a criterion ended up not being substantial to the end-result, the modelers would see this during the interactive spatial MCDA approach, and that itself would foster insightful discussions. We envisioned this being a very

collaborative, participatory, iterative process, not driven by only modeling results using one value.

*Section 3.3 – CBA*

Benefits are evaluated in terms of a fixed damage/hectare, without considering any specification for the soil use / exposed elements (residential, agriculture, industrial, …). Please, add some consideration about the accuracy and robustness of the used fixed value (=0.478 M$/he).

Response:    This fixed value is what was used within the USACE planning framework, which is what we are comparing against and recommending improving for decision-making. It is not common practice within the case-study area to assess flood reduction benefits in terms of their intended land use, although this could be an area of future study.

*Section 3.4 – Integrated CBA + MCA*

"Since the unique indicators contained different units of measurement ($/hectares, 0-100 risk) we used z-score normalization to transform the values to equivalent scales": this is not fully true. CB is non-dimensional (ranging 0-1 if benefits exceed costs, but later I understand expressed as 0-100); R and IF are also non-dimensional (ranging 0-100). Why was then the z-normalization used? This point should be clarified.

Response:    Per Nardo et al. (2005), z-score standardization is commonly used to convert all indicators to a common scale that averages zero with standard deviation (SD) of one. The average of zero helps reduce aggregation distortions of extremes. It is the standardization approach used for the popular Environmental Performance Index (EPI, https://epi.yale.edu/). In our $IF_{E|S}$ results shown in Table 4, note how the social impact factors are quite low (30-40s), while the environmental impacts are much higher (60-80s). This happened to be how the spread of social risks and environmental risks played out within our watershed and the zones of flood inundation, but in another case study area, we could see the opposite, all highs, all lows or any combination. By not z-scoring these values, we might have skewed the results toward being highly contingent on the environmental risks. Or, we might have resulted in many of the Alternatives that had a very high $CB_i$, similar to the Baseline condition of 100 (e.g., A3, A5, A8), because their costs were quite high compared to the level of reduced flood inundation achieved, then our resulting $T_i$ and ranking of Alternatives may have dis-favored some of these.

You are right about the rationale not being per the unit measurements. This reasoning has been clarified in Lines 343-344.

Moreover, the choice of the uniform weighting of the three indicators ($CB$, $IF_E$, $IF_S$) is an important part of the weight determination: what is the rationale under such choice? was it discussed with stakeholders? Was any sensitivity analysis performed?

Response:    See response to comment in Section 3.2.3. These weights are intended to be changed by the stakeholders, according to local values, and not per a calculated impact on the model sensitivity (which we find to be too prescriptive for our intended goals of participatory learning through the modeling exercise presented here).

I have a wider doubt here, about the consistency of T, presumably as a consequence of not having understood how factors IFs (or Rs) are formed. Let us take two mitigation alternatives ($A_1$, $A_2$), with different costs C, different benefits B, different impacts IF which, for simplicity, we here limit to the number of people affected by the flood (we call it N). Let us imagine that $C_1>C_2$ but also $B_1>B_2$ ($A_1$ reduces the flooded area more than $A_2$) so that $CB_1=CB_2$. Thus, the CB component of T will be equal for the two cases, thus suggesting that T is an indicator of efficiency (intensiveness) rather than of efficacy (extensiveness). Let us imagine that $N_2>N_1$ (as $A_1$ reduces the flooded are more than $A_2$): what would happen to $IF_1$ and $IF_2$? Is their contribution to T consistent with an indicator of efficiency? What for all the other components of $IF_E$ and $IF_S$?

Response:     What you are describing is precisely our rationale for presenting this study. In the above hypothetical example, where Alternative 1 was more expensive but also produced the greatest flood benefit, while Alternative 2 was less expensive with much less flood benefit, these two Cost-benefit ratios could theoretically be equal. In such a case, we propose that additional considerations of looking at the environmental and social risks and impacts in the greater region could shed more light on which alternatives are preferred by the local stakeholders, by viewing things holistically. The hypothetically-equal $CB_i$'s in this example would not change just because the social/environmental risk changed between the two Alternatives, *but the total risk ($T_i$) would change*, which is precisely our point. $T_i$ should be used for rankings rather than $CB_i$.

**Minor Points:**

Line 25:        "Hurricane Harvey": Add date of the event.

Response:     This sentence was removed in the revised manuscript. The year 2017 was added to the first reference of Hurricane Harvey in the new manuscript, Line 116. The exact dates are also included when referencing the NOAA rainfall data used in the hydrological model, Line 214-215 (and the Supplementary Information).

Line 150:      "Wealthy and middle-income populations face higher risks when located outside of federally-designated floodplains where flood insurance is voluntary". I cannot understand the reason of this higher risk.

Response:     Their risks due to being flooded are not higher, but their ability to re-build after the storm are inextricably higher when lacking flood insurance but a legal requirement to continue paying on a ~ 30-year mortgage for a home that is now un-sellable in its current condition. This sentence was removed in the revised manuscript, and additional clarification was added to describe homes within the federally-demarcated flood zone and those where flood insurance is voluntary (see response to comment for Line 271).

Comment:     Analyzed mitigation strategies (tab. 1) are those proposed in the USACE 2020 report. In particular, costs of the mitigation actions are derived from such reports. The reader would expect that also hydrological / hydraulic scenario are derived from the report but, apparently, it is not so, as modelling of such scenarios is discussed along the manuscript. Authors should clarify this point and discuss consistency of scenarios with associated costs. In particular, for scenario A7 I read (line 400) that the authors used a wider channel extension with respect to the USACE proposal: what about costs, were they modified accordingly?

Response:    The modeling is conducted in this study to showcase to the decision-makers, i.e. those that wrote and use such dam mitigation planning reports, how the spatial MCDA approach can be easily integrated into the approach those engineers/modelers already use, which is HEC-HMS/HEC-RAS modeling. Only the outcomes of the HEC-HMS/HEC-RAS models are described qualitatively in such reports. The models were re-created here for validity of the approach and to obtain the resulting spatial flood bounds for each of the Alternatives presented by the USACE.

Regarding the particulars of noting how a wider channel was actually necessary than what the USACE assumed – this is of more local concern than utility for the overall theme of this paper; therefore, it was moved to the Supplementary Information material. We only noted this here to showcase that the preferred USACE mitigation strategy, once it enters into detailed engineering design, will likely require much more land buy-out than was being considered; therefore, our inclusion of the USACE cost estimate was conservative. Cost estimates at this early planning stage typically always have large contingencies built in, so this is still a valid approach to showcase our framework.

Line 235:    "To standardize the point and polyline feature classes into spatially varied datasets, the Euclidean Distance method was applied. Euclidean distances convert feature layers into gridded datasets by assigning a value to each cell that indicates the distance of that cell to the nearest criterion, thus standardizing space and creating hotspots in multi-criteria decision making". Meaning and impact of this procedure is not clear to me.

Response:    A popular reference for this approach within GIS hotspot mapping was added to the manuscript (Line 262) in case the reader would like to learn more about the Euclidean distance. This is a very common approach in spatial MCDA (e.g., below references, for a few examples), and is appropriate to list here.

Dell'Ovo, M., Capolongo, S., & Oppio, A. (2018). Combining spatial analysis with MCDA for the siting of healthcare facilities. *Land Use Policy*, *76*, 634-644.

Rufino, I. A GIS-based Multi-Criteria Analysis (MCDA) approach for water shortage risk mapping.

Demesouka, O. E., Vavatsikos, A. P., & Anagnostopoulos, K. P. (2013). Suitability analysis for siting MSW landfills and its multicriteria spatial decision support system: method, implementation and case study. *Waste management*, *33*(5), 1190-1206.

Tammi, I., & Kalliola, R. (2014). Spatial MCDA in marine planning: Experiences from the Mediterranean and Baltic Seas. *Marine Policy*, *48*, 73-83.

Aşilioğlu, F. (2021). GISimos MCDA land suitability model for ecotourism development. *Journal of Environmental Engineering and Landscape Management*, *29*(3), 200-214.

Cetinkaya, C., Özceylan, E., Erbaş, M., & Kabak, M. (2016). GIS-based fuzzy MCDA approach for siting refugee camp: A case study for southeastern Turkey. *International Journal of Disaster Risk Reduction*, *18*, 218-231.

| Line 271: | "The spatial risk associated with flood insurance was derived from national flood hazard zones and a repository of damaged structures in the community. It was assumed that residents within the FEMA 1% and 0.1% flood zones carried flood insurance, while 20% of all other residents had purchased voluntary insurance". Please, provide comment about soundness of such assumption. |
|---|---|
| Response: | Flood insurance is mandatory within the FEMA 1% and 0.1% flood zones. Persons cannot have a mortgage nor regular home insurance without also carrying FEMA flood insurance if they reside within one of these zones. Per Klotzbach et al. (2018), less than 20% of all damaged homes within Harris County impacted by Hurricane Harvey possessed active flood insurance, as many homes were located where insurance is voluntary. This reference has been clarified in Lines 171-173 and 310-311. |
| Table 3: | I cannot reconstruct values for CB. Let us take alternative A3 for Addicks as an example. Reduction of flooded area is 466 he; when multiplied by 0.478 M$/he we obtain a damage reduction B = 223 M$; cost is here C = 5000 M$ (please, make indication of units coherent in tables 1 and 3) with a consequent CB = C/B = 2200% … not comparable with values in table 3 and with common sense. This clearly means that there is something I have not understood (I honestly tried) … please, clarify. |
| Response: | Indication of units was added to the table (now Table 4) in revised manuscript. |
| | In the above calculations, the benefit-cost ratio is being calculated instead of the cost-benefit ratio. Rather, the cost-benefit ratio is the inverse of the benefit-cost ratio. In the A3 example, Benefit over Cost = 223 M$ / 5000 M$ = 0.045. Therefore, Cost over Benefit = 1 − 0.045 = 0.955 = 95.5%. |
| Figure 6: | I cannot understand spider graphs (c) and (d); I expect the same values as in plots (a) and (b) to be represented, but there is no coherence. |
| Response: | (a) and (c) should correspond, as well as (b) and (d). This figure and caption have been updated to remove the spider graphs, since they are repetitive. |

**Suggestions on Organization of Manuscript:**

| Suggestion 1: | Section 2.1: This relatively long paragraph appears as a continuation/specification of the literature review provided in the Introduction rather than a description in the methodology here used. Moreover, the scheme for "integrated flood management decision-making" in Fig. 1 is here presented but not really used (or, perhaps, not clearly explained). Consider better focusing all information within lines 18-104 with respect to the specific aim of the paper (case study). |
|---|---|
| Response: | The introduction and background sections have been re-written to condense the information and highlight the paper goals. The introduction to the case study was streamlined within Section 1 and Section 2.1. Due to the significant inter-disciplinary impacts occurring within this watershed across three scientific domains, the background section was maintained but was re-written and re-organized to include only the information necessary for introducing the reader to the study and the alternative |

mitigation strategies for adequate understanding and reproducibility when reading the Methodology and Results sections.

Suggestion 2:  Along the manuscript discussions are alternated between the hydrological/hydraulic scenarios (Sections 2.2.1, 3.1, 4.1) and the impact/damage/cost scenarios (remaining sections). I suggest to re-organize the material so that the sections for the two groups are presented all together, and the flux of information should become more coherent. This would also avoid some repetitions.

Response:  The methodologies for the flood modeling, CBA, and spatial MCDA are unique and are maintained in separate sections; for clarity, additional sentences are described in Section 3 (Lines 199-205) to amalgamate these themes prior to reading.

The results of the approaches are integrated more thoroughly in Section 4. Also, much of the extra information regarding details of the flood modeling outputs are moved to the Supplementary Information documentation to reduce manuscript size and limit the discussion of results to what is pertinent for understanding how the combined CBA+MCDA could alter decision-making.

Suggestion 3:  Lines 314-319: Here a general discussion about principles of a CBA is provided. However, here a simplified version of benefits evaluation is used. The general discussion could be omitted or moved to some introductory section.

Response:  We are unsure what is referenced by a 'simplified version of benefits evaluation'. The cited articles (i.e., Ward, 2012 and Jonkman et al., 2004) describe how benefits, in the realm of water resources policy making, are informed by trade-offs in hydrological efficacy of the intended structural mitigation solution. Here, we used spatial trade-offs regarding flood inundation extent, which is what was also used in the USACE planning mitigation studies referenced in the Manuscript and here in the responses. This was to showcase how our integrated CBA+MCDA approach can fit within standard practicing planning paradigms. If the reviewer comment means that benefits evaluations can be applied in a more robust fashion, such as by assessing additional hydrological improvements amongst each of the alternatives (i.e., peak flow reduction, time to peak, etc.?), we chose the benefit of spatial flood risk due to its popularity within flood mitigation MCDA studies and within practicing dam infrastructure mitigation reports.

We have moved these sentences to the Introduction section, Lines 78-82, near the background discussion about MCDA to respond to the reviewer comment.